# Constitutive scaffolding of multiple Wnt enhanceosome components by Legless/BCL9

**Laurens M van Tienen, Juliusz Mieszczanek, Marc Fiedler, Trevor J Rutherford, Mariann Bienz***

MRC Laboratory of Molecular Biology, Cambridge, United Kingdom

**Abstract** Wnt/$\beta$-catenin signaling elicits context-dependent transcription switches that determine normal development and oncogenesis. These are mediated by the Wnt enhanceosome, a multiprotein complex binding to the Pygo chromatin reader and acting through TCF/LEF-responsive enhancers. Pygo renders this complex Wnt-responsive, by capturing $\beta$-catenin via the Legless/BCL9 adaptor. We used CRISPR/Cas9 genome engineering of *Drosophila legless (lgs)* and human *BCL9* and *B9L* to show that the C-terminus downstream of their adaptor elements is crucial for Wnt responses. BioID proximity labeling revealed that BCL9 and B9L, like PYGO2, are constitutive components of the Wnt enhanceosome. Wnt-dependent docking of $\beta$-catenin to the enhanceosome apparently causes a rearrangement that apposes the BCL9/B9L C-terminus to TCF. This C-terminus binds to the Groucho/TLE co-repressor, and also to the Chip/LDB1-SSDP enhanceosome core complex via an evolutionary conserved element. An unexpected link between BCL9/B9L, PYGO2 and nuclear co-receptor complexes suggests that these $\beta$-catenin co-factors may coordinate Wnt and nuclear hormone responses.

***For correspondence:** mb2@mrc-lmb.cam.ac.uk

**Competing interests:** The authors declare that no competing interests exist.

## Introduction

The Wnt/$\beta$-catenin signaling cascade is an ancient cell communication pathway that operates context-dependent transcriptional switches to control animal development and tissue homeostasis (*Cadigan and Nusse, 1997*). Deregulation of the pathway in adult tissues can lead to many different cancers, most notably colorectal cancer (*Clevers and Nusse, 2012*). Wnt-induced transcription is mediated by T cell factors (TCF1/3/4, LEF1) bound to Wnt-responsive enhancers, but their activity depends on the co-activator $\beta$-catenin (Armadillo in *Drosophila*), which is rapidly degraded in unstimulated cells. Absence of $\beta$-catenin thus defines the OFF state of these enhancers, which are silenced by Groucho/TLE co-repressors bound to TCF via their Q domain. This domain tetramerizes to promote transcriptional repression (*Chodaparambil et al., 2014*), which leads to chromatin compaction (*Sekiya and Zaret, 2007*) apparently assisted by the interaction between Groucho/TLE and histone deacetylases (HDACs) (*Jennings et al., 2008*; *Turki-Judeh and Courey, 2012*).

Wnt signaling relieves this repression by blocking the degradation of $\beta$-catenin, which thus accumulates and binds to TCF, converting the Wnt-responsive enhancers into the ON state. This involves the binding of $\beta$-catenin to various transcriptional co-activators via its C-terminus, most notably to the CREB-binding protein (CBP) histone acetyltransferase or its p300 paralog (*Mosimann et al., 2009*), resulting in the transcription of the linked Wnt target genes. Subsequent reversion to the OFF state (for example, by negative feedback from high Wnt signaling levels near Wnt-producing cells, or upon cessation of signaling) involves Groucho/TLE-dependent silencing, but also requires the Osa/ARID1 subunit of the BAF (also known as SWI/SNF) chromatin remodeling complex (*Collins and Treisman, 2000*) which binds to $\beta$-catenin through its BRG/BRM subunit (*Barker et al.,*

**eLife digest** In every animal, different cells must be able to communicate with each other to make sure that the body is correctly formed and maintained. Animal cells have many ways of communicating, but one important and well-studied mechanism involves a signaling molecule called Wnt that is released by some cells and received by others. The Wnt molecule and its effects are similar in all animals, and over-active Wnt signaling in humans contributes to a number of diseases including various cancers.

The Wnt signal is carried from the surface of the receiving cell to the DNA in its nucleus via a protein called $\beta$-catenin. The $\beta$-catenin protein then helps to switch on a large number of genes. However, to do this $\beta$-catenin must interact with an assembly of other proteins collectively called the Wnt enhanceosome. There are still many unknowns about how exactly $\beta$-catenin cooperates with the enhanceosome.

Now, van Tienen et al. investigated one component of the Wnt/$\beta$-catenin pathway called BCL9/B9L: a large protein that contains a number of flexible regions. First, a biochemical technique called BioID was used with human embryonic kidney cells to determine the proteins that BCL9/B9L encounters during a 12-hour period. This technique can detect when two proteins come close together, even if the interaction is weak or does not last very long.

The BioID experiments showed that BCL9/B9L is close to two proteins in the Wnt enhanceosome in addition to $\beta$-catenin, and other techniques were used to confirm that one of these proteins contacts BCL9/B9L directly. The experiments also showed that BCL9/B9L acted as a tether to bring $\beta$-catenin close to the protein within the enhanceosome that binds to the DNA. Importantly, BCL9/B9L interacted with the enhanceosome both in the presence and absence of Wnt, indicating that the assembly is ready to switch genes on as soon as $\beta$-catenin reaches the DNA.

Next, van Tienen et al. confirmed that the parts of BCL9/B9L that bind to the enhanceosome are important for its activity by using a gene-editing technology called CRISPR/Cas9 to essentially delete them both in the human cells and in fruit flies.

Unexpectedly, the BioID experiments also revealed that BCL9/B9L binds to proteins that transmit signals from molecules other than Wnt, in particular from hormones such as estrogen and androgen. Future experiments could explore if, and how, BCL9/B9L integrates these signals from hormones with the signal from Wnt. A better understanding of this process might have important implications for the treatment of certain cancers, such as breast and prostate cancers that can be driven by over-active hormone signals.

*2001*). Cancer genome sequencing has uncovered a widespread tumor suppressor role of the BAF complex, which guards against numerous cancers including colorectal cancer, with >20% of all cancers exhibiting at least one inactivating mutation in one of its subunits, most notably in ARID1A (*Kadoch and Crabtree, 2015*). Thus, it appears that failure of Wnt-inducible enhancers to respond to negative feedback imposed by the BAF complex strongly predisposes to cancer.

How $\beta$-catenin overcomes Groucho/TLE-dependent repression remains unclear, especially since $\beta$-catenin and TLE bind to TCF simultaneously (*Chodaparambil et al., 2014*). Therefore, the simplest model envisaging competition between $\beta$-catenin and TLE cannot explain this switch, which implies that co-factors are required. One of these is Pygo, a chromatin reader binding to histone H3 tail methylated at lysine 4 (H3K4m) via its C-terminal PHD finger (*Fiedler et al., 2008*). In *Drosophila* where Pygo was discovered as an essential co-factor for activated Armadillo, its main function appears to be to assist Armadillo in overcoming Groucho-dependent repression (*Mieszczanek et al., 2008*). We recently discovered that Pygo associates with TCF enhancers via its highly conserved N-terminal NPF motif that binds directly to the ChiLS complex, composed of a dimer of Chip/LDB (LIM domain-binding protein) and a tetramer of SSDP (single-stranded DNA-binding protein, also known as SSBP). Notably, ChiLS also binds to other enhancer-bound NPF factors such as Osa/ARID1 and RUNX, and to the C-terminal WD40 domain of Groucho/TLE, and thus forms the core module of a multiprotein complex termed 'Wnt enhanceosome' (*Fiedler et al., 2015*). We proposed that Pygo renders this complex Wnt-responsive by capturing Armadillo/$\beta$-

catenin through the Legless adaptor (whose orthologs in humans are BCL9 and B9L, also known as BCL9-2) (*Kramps et al., 2002*; *Städeli and Basler, 2005*). The salient feature of our model is that the Wnt enhanceosome keeps TCF target genes repressed prior to Wnt signaling while at the same time priming them for subsequent Wnt induction, and for timely shut-down via negative feedback depending on Osa/ARID1 (*Fiedler et al., 2015*).

Here, we assess the function of Legless and BCL9/B9L within the Wnt enhanceosome. Using a proximity-labeling proteomics approach (called BioID; *Roux et al., 2012*) in human embryonic kidney (HEK293) cells, we uncovered a compelling association between BCL9/B9L and the core Wnt enhanceosome components, regardless of Wnt signaling. Co-immunoprecipitation (coIP) and in vitro binding assays based on Nuclear Magnetic Resonance (NMR) revealed that BCL9 and B9L associate with TLE3 through their C-termini, and that they bind directly to ChiLS via their evolutionary conserved homology domain 3 (HD3). These elements are outside the sequences mediating the adaptor function between Pygo and Armadillo/$\beta$-catenin, but they are similarly important for Wnt responses during *Drosophila* development and in human cells, as we show by CRISPR/Cas9-based genome editing. Our results consolidate and refine the Wnt enhanceosome model, indicating a constitutive scaffolding function of BCL9/B9L within this complex. Our evidence further suggests that BCL9/B9L but not Pygo undergoes a $\beta$-catenin-dependent rearrangement within the enhanceosome upon Wnt signaling, gaining proximity to TCF, which might trigger enhanceosome switching.

## Results

### The C-terminus of Legless is required for Wingless signaling during fly development

Legless and BCL9/B9L paralogs (collectively referred to as Legless/BCL9 below) bind to Pygo and Armadillo/$\beta$-catenin via their conserved N-terminal homology domains 1 and 2 (HD1, HD2), respectively (*Figure 1A*). Previous evidence suggested that the sole function of Legless during *Drosophila* development is to link Pygo to Armadillo (*Kramps et al., 2002*). However, C-terminal truncations of BCL9/B9L behave as dominant-negatives regarding Wnt responses in mammalian cell lines (*Adachi et al., 2004*; *de la Roche et al., 2008*), which suggested that their C-termini harbor functionally relevant elements. Indeed, these elements are missing in a Legless truncation encoded by a known *lgs* mutation (*lgs^7l^*), owing to a stop codon downstream of HD2 (*Kramps et al., 2002*), yet this truncation should be fully competent to link Pygo to Armadillo. Consistent with the notion of functional elements downstream of HD2, the C-termini of Legless/BCL9 exhibit several conserved sequence blocks, in particular HD3 (*Figure 1A*) which is found in all orthologs from the most primitive animals to humans. No ligand has been identified for HD3, nor for the C-terminus of Legless/BCL9.

To test the function of these sequences downstream of HD2, we deleted HD3 from endogenous *Drosophila lgs* (*lgs^ΔHD3^*) using the CRISPR/Cas9 system (*Port et al., 2014*). We also generated a truncation allele (*lgs^2-8^*) that deletes HD3 plus the entire C-terminus (*Figure 1—figure supplement 1*). We isolated fly lines bearing these alleles, and confirmed that they express mutant proteins of the predicted sizes, and at normal levels (*Figure 1B*). Notably, the *lgs^2-8^* truncation allele causes pupal lethality if combined with a strong *lgs* allele (*lgs^20F^*; *Kramps et al., 2002*). Furthermore, homozygous *lgs^2-8^* animals (derived from homozygous mothers) show severely delayed development, and most die in their pupal cases, with <25% of them hatching as flies (*Figure 1C*). A high fraction of these escapers exhibit patterning defects indicative of attenuated signaling by Wingless (Wg, *Drosophila* Wnt), as previously observed in flies bearing weak alleles of *lgs*, *pygo*, and *dTCF* (*Brunner et al., 1997*; *Kramps et al., 2002*; *Thompson et al., 2002*)—namely proximal leg duplications and dorso-ventral polarity leg defects (in ~1/2 mutant flies) as well as loss of sternites in the ventral abdomen (in ~1/6 mutant flies; *Figure 1D*). Identical phenotypes were observed in homozygotes bearing an equivalent truncation allele isolated independently (*lgs^1-5^*; *Figure 1—figure supplement 1*), ruling them out as off-target effects of the CRISPR/Cas9 engineering process. Thus, the sequences downstream of HD2 are essential for *Drosophila* development, and appear to participate in Wg signaling responses.

To confirm this, we examined a Wg-responsive transcriptional enhancer from *dpp* (*dpp.lacZ*; *Blackman et al., 1991*), which is repressed by high levels of Wg signaling emanating from the Wg

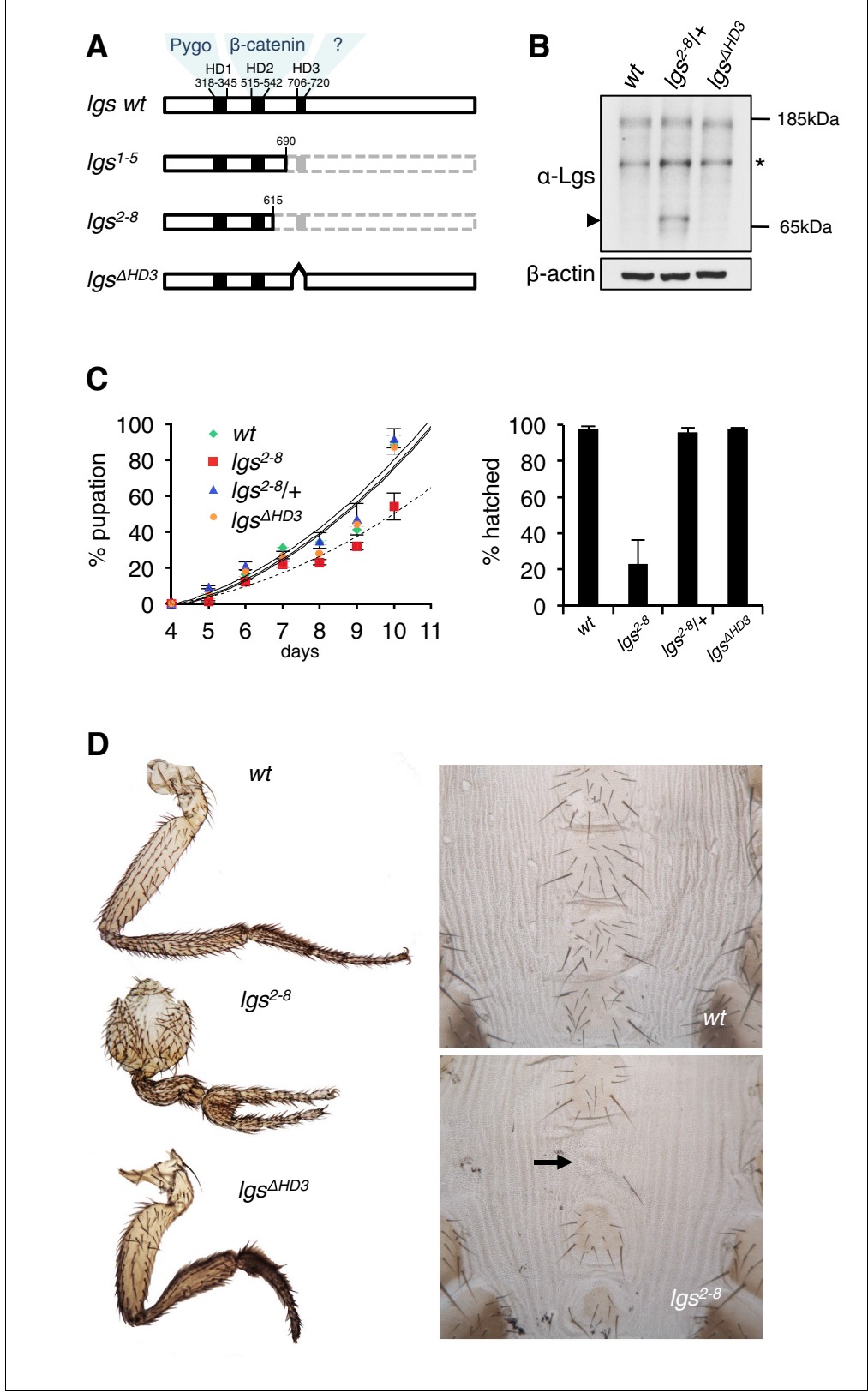

**Figure 1.** The C-terminus of Legless is required for Wg-dependent patterns in flies. (**A**) Cartoon of *lgs* mutants, with domain boundaries indicated (*grey*, deleted sequences). (**B**) Western blot of lysates from *lgs* mutant embryos (genotypes indicated above panels), probed with antibodies as indicated, confirming stability of the *lgs²⁻⁸*

*Figure 1 continued on next page*

*Figure 1 continued*

truncation product (~65 kDa, *arrowhead*; an unspecific cross-reactivity of this α-Lgs antiserum is marked by *asterisk*). (C) Developmental rates and survival of wt and *lgs* homozygous mutant larvae as indicated; first-instar larvae were picked (n = 25), and % pupation (*left*) or hatching of flies (*right*) was scored daily; error bars, SEM of four independent experiments. (D) Posterior leg and abdominal phenotypes of wt and *lgs* mutant flies, as indicated; representative examples are shown; *arrow*, missing sternite.

The following figure supplement is available for figure 1:

**Figure supplement 1.** CRISPR/Cas9-based gene editing strategies for *Drosophila legless*.

---

source in ventral compartments of leg discs via a homeodomain protein called Brinker (*Theisen et al., 2007*). Immunofluorescence revealed a striking derepression of *dpp.lacZ* in the ventral compartment of >1/3 of the leg discs from *lgs*$^{2-8}$ homozygotes (28/78 discs) (*Figure 2A,B*; *Figure 2—figure supplement 1*), demonstrating that the function of the mutant Legless in transducing the Wg signal is severely compromised in these discs. This was also true for wing discs (dissected from of pupating larvae, to ensure matched developmental timing between mutants and wild-type, wt): these discs exhibit much reduced expression of *Senseless* (*Sens*) (n = 50 discs; *Figure 2C,D*), a Wg target gene that specifies bristles along the wing margin and whose expression is abolished in *pygo* mutant clones (*Parker et al., 2002*; *Fiedler et al., 2008*). As expected, this attenuation of Sens results in fewer margin bristles in the homozygous mutant escaper flies (61 versus 80 stout margin bristles, and 15.6 versus 18.8 chemosensory bristles, per wt versus mutant wing, respectively), and larger gaps between individual stout bristles (22.7 versus 17.3 µm, per mutant versus wt wing, respectively) (*Figure 2E,F*; mean values from 10 wings dissected from different homozygous *lgs*$^{2-8}$ females; the sizes of wt and mutant wings were the same).

We also used RT-qPCR to examine RNA expression levels of the endogenous Wg target genes *engrailed*, *Distalless* and *H15* (*Cadigan and Nusse, 1997*; *Estella et al., 2008*; *Wilder and Perrimon, 1995*) in lysates from wing discs dissected from climbing (that is, fully grown) homozygous *lgs*$^{2-8}$ larvae. We found significant reductions in the expression levels of all three target genes compared to controls (*Figure 2G*; note that neither *dpp* nor *dpp.lacZ* expression is controlled by Wg in wings discs; see *Figure 2C,D*). Finally, we also found that heterozygosity of *lgs*$^{2-8}$ suppresses the rough eye phenotype caused by activated Armadillo (*Freeman and Bienz, 2001*), indistinguishably from heterozygosity of *pygo*$^{S123}$ or *lgs*$^{20F}$ (*Thompson et al., 2002*; *Kramps et al., 2002*) (*Figure 2—figure supplement 1*).

These results demonstrate that the C-terminal Legless truncation encoded by our *lgs*$^{2-8}$ allele is severely compromised in its ability to transduce the Wg signal in multiple developmental contexts, despite retaining normal adaptor function in linking Pygo and Armadillo. Evidently, this adaptor function is not sufficient to sustain normal development if Legless is expressed at endogenous levels, suggesting that the previously reported rescue activities of equivalent Legless fragments (*Kramps et al., 2002*) may have stemmed from overexpression.

We also examined homozygous *lgs*$^{ΔHD3}$ animals, which hatched as flies without showing any developmental delay (*Figure 1C*). However, ~5% of these flies exhibited leg abnormalities, similarly to those exhibited by *lgs*$^{2-8}$ (*Figure 1D*) although the penetrance of this phenotype was much higher in the latter (see above). Nevertheless, these leg defects in the *lgs*$^{ΔHD3}$ homozygotes suggest that HD3 contributes to the function of Legless in transducing Wg responses.

## The C-terminus of BCL9/B9L is required for Wnt responses in human cells

Given these results from flies, we decided to also assess the function of the BCL9/B9L C-terminus and HD3 in the human Wnt response. We thus applied the CRISPR/Cas9 system to HEK293T cells (*Ran et al., 2013*), to generate single knock-out (KO) cells lacking BCL9 or its nuclear paralog B9L, or double knock-out (DKO) cells lacking both (*Figure 3—figure supplement 1*). Using a TCF-dependent reporter assay (called SuperTOP; *Veeman et al., 2003*), we found the Wnt-induced transcriptional activity of these DKO cells to be substantially reduced (to <20% of their parental control cells; *Figure 3A*). Loss of BCL9 reduces the transcriptional activity nearly as much as loss of both paralogs,

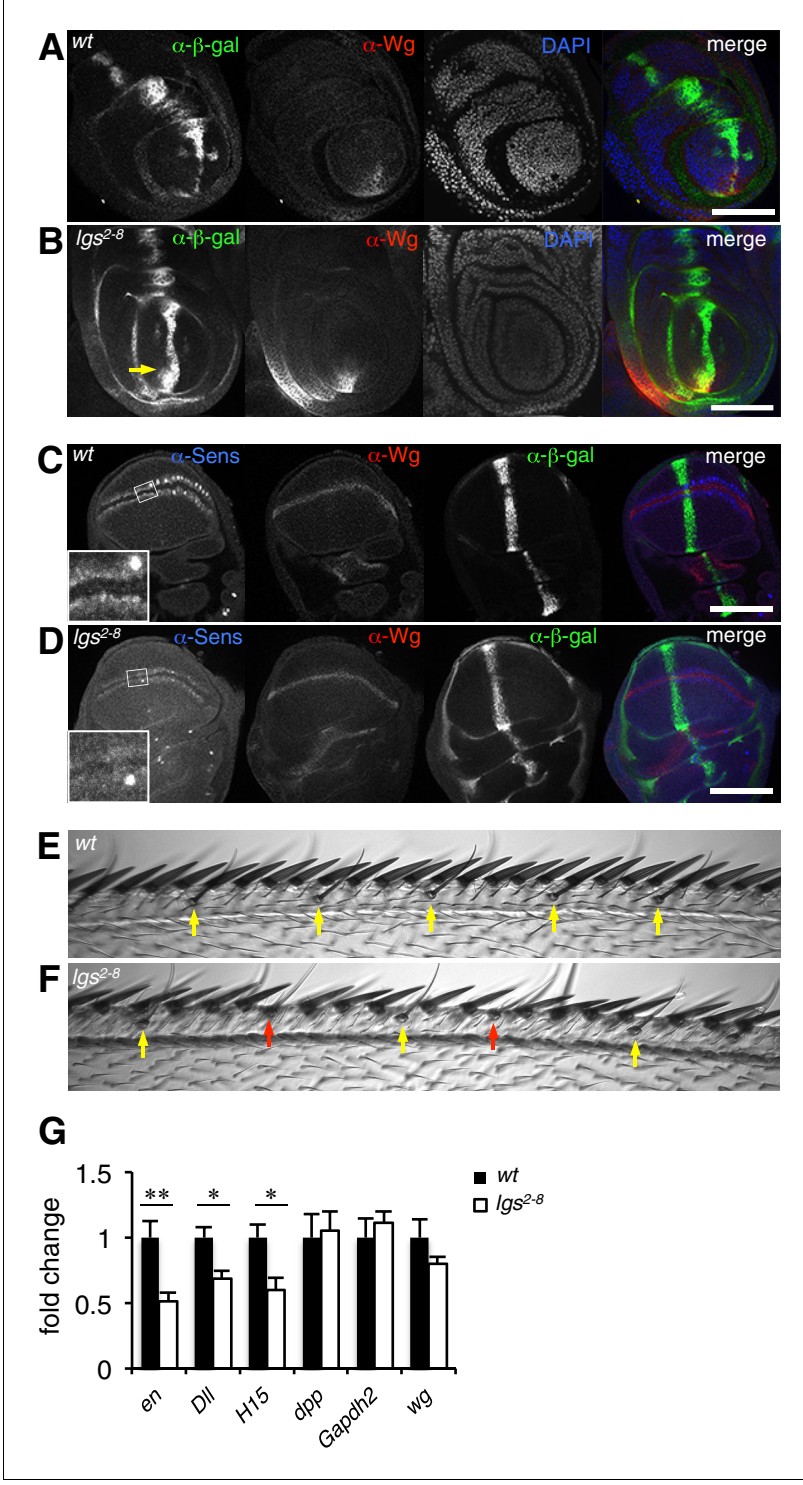

**Figure 2.** The C-terminus of Legless is required for nuclear Wg responses in imaginal discs. (**A**, **B**) Third leg discs from third instar (**A**) wt or (**B**) *lgs²⁻⁸/lgs²⁻⁸* larvae, fixed and stained with antibodies as indicated in panels (merges on the right), showing derepression of *dpp.lacZ* in the ventral compartment (arrow); space bar, 50 µm. (**C**, **D**) Corresponding wing discs, showing attenuated Sens expression along the prospective wing margin (see also insets); space bar, 100 µm. (**E**, **F**) Anterior margin segments of escaper flies, focused on stout margin bristles; *yellow arrows*, chemosensory bristles; *red arrows*, missing stout bristles causing gaps that are occupied by ectopic chemosensory bristles. (**G**) RT-qPCR assays of wing discs dissected from climbing wt and *lgs²⁻⁸/lgs²⁻⁸* third instar

*Figure 2 continued*
larvae, as indicated; values were normalized relative to *RpL32* (internal control), and are shown as mean ± SEM
relative to wt (set to 1); * = p<0.05, ** = p<0.01.
The following figure supplement is available for figure 2:

**Figure supplement 1.** Additional analysis of *lgs²⁻⁸* during fly development.

suggesting that BCL9 may be the physiologically limiting paralog in these cells, at least for transient
Wnt responses. Likewise, the endogenous Wnt target genes *AXIN2* (*Lustig et al., 2002*) and *SP5*
(*Hoverter et al., 2012*; *Weidinger et al., 2005*) are no longer Wnt-inducible in the DKO cells
(*Figure 3B*), reconfirming the critical role of BCL9/B9L in the Wnt response of these cells.

When we re-expressed FLAG-B9L in the DKO cells, this restored full Wnt-responsiveness
(*Figure 3C*). However, the C-terminal truncation mutants of B9L (△C, △HD3△C) failed to do so
(despite elevated expression levels in the case of △HD3△C), and the HD3 deletion provided only
partial rescue activity, similarly to a deletion mutant lacking the HD1 Pygo-binding element

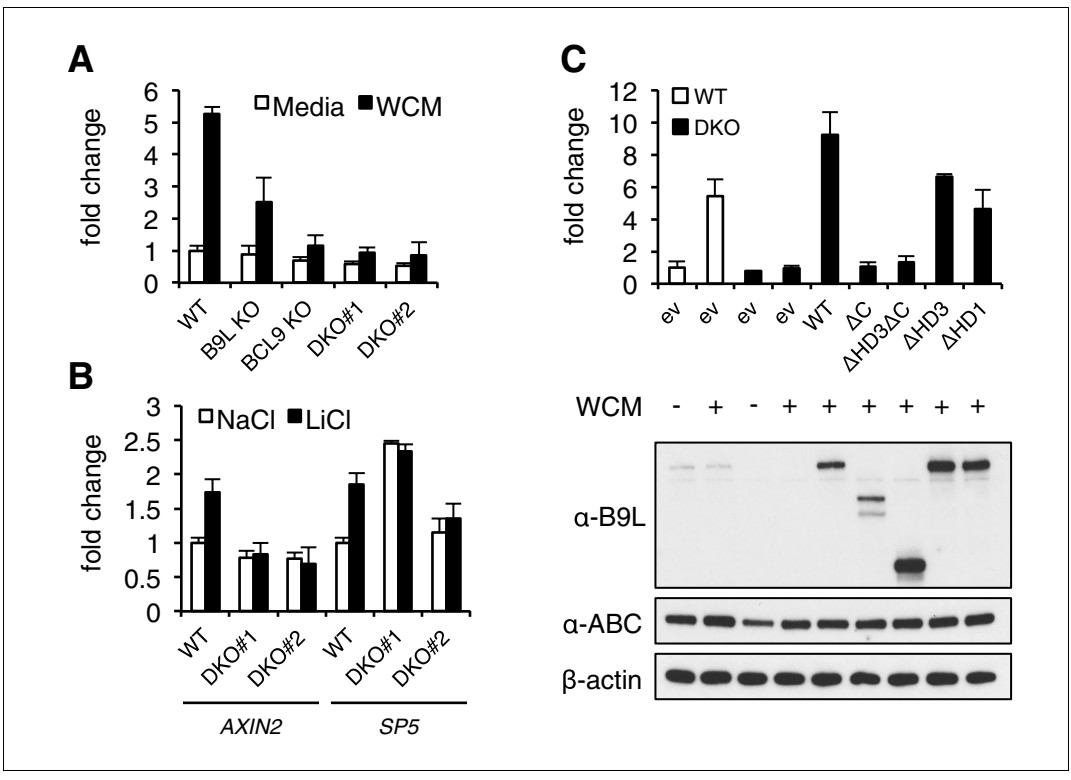

**Figure 3.** The C-terminus of BCL9/B9L is required for transcriptional Wnt responses in human cells. (**A**) SuperTOP
assays in wt or KO HEK293T cells lacking BCL9 and/or B9L, as indicated, ±6 hr of Wnt stimulation (WCM, Wnt3a-
conditioned-media); mean ± SEM (n = 3 independent experiments). (**B**) RT-qPCR assays in wt or DKO HEK293T
cells, ±6 hr of 20 mM NaCl or LiCl as indicated, revealing relative transcript levels of *AXIN2* or *SP5*; values were
normalized to *TBP* (internal control), and are shown as mean ± SEM relative to unstimulated wt controls (set to 1);
note that the uninduced levels of *SP5* were unusually high in one of the two DKO clones, but neither of the DKO
clones showed Wnt-inducible *SP5*. (**C**) SuperTOP assays in wt or DKO HEK293T cells as in (**A**), 24 hr after
transfection with wt and mutant B9L as indicated (*below*, corresponding Western blots; α-ABC, active
unphosphorylated *β*-catenin); ev, empty vector control.
The following figure supplement is available for figure 3:

**Figure supplement 1.** CRISPR/Cas9-based gene editing strategies for *BCL9* and *B9L* in HEK293T cells.

(*Figure 3C*). This demonstrates the crucial role of the B9L C-terminus for the Wnt response of these cells, and it indicates a functional contribution of HD3 to this response. Restoration of Wnt-responsiveness is also provided by re-expressed BCL9 tagged with green fluorescent protein (GFP-BCL9) albeit not by its C-terminal truncation, but we did not analyze this paralog further since its transcriptional activation potential is not as strong as that of the nuclear B9L (*de la Roche et al., 2008*).

## B9L and BCL9 are constitutively associated with the Wnt enhanceosome

Given the requirement of the BCL9/B9L C-terminus for the Wnt response, we set out to identify its ligands that confer this function. Since previous attempts based on tandem-affinity pull-down experiments were unsuccessful (M Graeb, PhD thesis), we adopted a proximity-labeling approach called BioID, tagging B9L and BCL9 with BirA* (a promiscuous version of the biotin ligase BirA; *Roux et al., 2012*) and using these as baits to probe the proteome associated with BCL9/B9L in cells. This method is capable of identifying transient low-affinity ligands of the bait as well as indirect bystanders (that is, indirect interactors and 'vicinal' proteins), provided these are within range of the BirA* tag: in the case of Lamin A, ~50% of the hits were estimated to be within 20–30 nm of its BirA* tag (*Roux et al., 2012*). The upper limit appears to be <100 nm, according to BioID studies with Cep250, an extended coiled-coil protein that spans ~100 nm within the centrosomal complex (as determined by super-resolution microscopy; *Sonnen et al., 2012*): the C-terminal ligand of Cep250 was only labeled by its C-terminal but not its N-terminal BirA* tag (*Firat-Karalar et al., 2014*). Therefore, the BioID approach can provide insight into the position and reach of a bait's BirA* tag within a protein complex. Note that BCL9/B9L is presumed to be an extended protein, with >90% of its sequence predicted to be intrinsically unstructured.

To keep the expression of our BCL9 and B9L baits near endogenous levels, we chose a tetracycline-controlled transcriptional activation system based on T-REx-293 cells (*Al-Jassar et al., 2017*), isolating clonal T-REx-293 cell lines that express B9L-BirA* and BCL9-BirA* integrated at a specific genomic locus. Cells were labeled with biotin for 12 hr, with or without stimulation by Wnt3A-conditioned media or 50 mM LiCl (to inhibit glycogen synthase kinase 3, which causes activation of β-catenin, and thus mimics Wnt stimulation). Lysates were prepared for one-step biotin-avidin affinity purification and subsequent analysis by LC-MS/MS mass spectrometry. Recall that the probability of identifying a hit by this approach (reflected by the total number of unweighted spectral counts derived from this hit) is primarily determined by its proximity to the bait, but is also affected by the strength and duration of their interaction during the labeling period, and by other factors including size and cellular abundance of interactors and suitably exposed biotin-acceptor sites (primary amines).

We first conducted several experiments with B9L-BirA* as a bait since the B9L paralog is predominantly nuclear, and thus likely dedicated to the transcriptional Wnt response, while BCL9 is distributed throughout the nucleus and cytoplasm (*de la Roche et al., 2008*) (*Figure 4—figure supplement 1*), possibly carrying out additional functions in the cytoplasm such as chaperoning β-catenin (*de la Roche et al., 2012*). Strikingly, each experiment identified known components of the Wnt enhanceosome, even in the absence of Wnt stimulation: we consistently found ARID1A and ARID1B as the top hits, followed closely by TLE3, TLE1, TLE4, LDB1 and PYGO2 and, further down the list, TCF factors, β-catenin and SSBP3/4 (initially named SSDP1/2; the peptides obtained do not distinguish between these closely related paralogs) (*Figure 4A*). We also found nine additional subunits of the BAF complex (*Kadoch and Crabtree, 2015*) on our list, topped by BRG-1/SMARCA4, with only BAF155/SMARCC1, BAF47/SMARCB1, SS18 and BCL7 missing (the latter three likely for technical reasons; *Figure 4—figure supplement 2*). This indicates that the fully assembled BAF complex is tethered to the Wnt enhanceosome, presumably through its enhancer-binding Osa/ARID1 subunit which also binds to ChiLS (see Introduction). Furthermore, we found the CBP co-activator and its p300 paralog, as well as DNA-binding proteins of the LHX and GATA families known to tether ChiLS to DNA (*Love et al., 2014*). Essentially the same set of proteins were found with BCL9-BirA* as a bait, albeit with lower efficiency (*Figure 4B*), possibly because of its largely cytoplasmic localization. Indeed, we found several cytoplasmic proteins specifically associated with BCL9 but not B9L, including α-catenin (*Figure 4—figure supplement 2*).

To determine whether any of these hits might be direct ligands of B9L, we applied the RIME (rapid IP mass spectrometry of endogenous proteins) technique (*Mohammed et al., 2016*) to our

**A**

| BioID | - Wnt | | + Wnt | | + LiCl | |
|---|---|---|---|---|---|---|
| Protein ID | WT | Ctrl | WT | Ctrl | WT | Ctrl |
| B9L (BAIT) | 386 | 0 | 419 | 0 | 320 | 0 |
| ARID1A | 203 | 1 | 365 | 0 | 372 | 1 |
| ARID1B | 109 | 0 | 277 | 5 | 300 | 0 |
| NCOR1 | 104 | 14 | 305 | 0 | 269 | 11 |
| NCOA2 | 88 | 0 | 143 | 0 | 134 | 0 |
| TLE3 | 65 | 2 | 159 | 1 | 167 | 1 |
| NCOR2 | 58 | 0 | 227 | 0 | 200 | 0 |
| SMARCA4 | 41 | 1 | 102 | 6 | 79 | 10 |
| TLE1 | 34 | 1 | 122 | 1 | 129 | 1 |
| QSER1 | 33 | 0 | 77 | 0 | 60 | 0 |
| RBM14 | 32 | 0 | 67 | 0 | 63 | 0 |
| GSE1 | 31 | 1 | 165 | 2 | 97 | 6 |
| NCOA3 | 31 | 0 | 105 | 0 | 99 | 0 |
| ZNF609 | 29 | 1 | 139 | 0 | 86 | 0 |
| TLE4 | 28 | 1 | 95 | 1 | 96 | 1 |
| BAHCC1 | 19 | 0 | 34 | 0 | 21 | 0 |
| NCOA6 | 17 | 1 | 57 | 0 | 61 | 5 |
| P300 | 16 | 0 | 70 | 1 | 57 | 1 |
| PYGO2 | 14 | 0 | 30 | 0 | 39 | 0 |
| ARNT | 13 | 0 | 32 | 1 | 14 | 0 |
| LDB1 | 13 | 0 | 17 | 0 | 13 | 0 |
| CBP | 12 | 0 | 67 | 1 | 64 | 0 |
| NCOA1 | 12 | 0 | 54 | 0 | 37 | 0 |
| SMARCA2 | 12 | 0 | 43 | 2 | 20 | 5 |
| GATA4 | 10 | 0 | 11 | 0 | 7 | 1 |
| TCF4 | 6 | 1 | 88 | 0 | 90 | 2 |
| ARID5B | 3 | 0 | 95 | 0 | 61 | 0 |
| AHR | 3 | 0 | 19 | 0 | 2 | 0 |
| LMX1B | 2 | 0 | 18 | 0 | 6 | 0 |
| GATA6 | 2 | 0 | 6 | 0 | 9 | 1 |
| SSBP3/4 | 1 | 0 | 6 | 0 | 8 | 0 |
| β-catenin | 0 | 0 | 48 | 0 | 11 | 2 |
| TCF3 | 0 | 0 | 30 | 0 | 34 | 2 |
| TCF1 | 0 | 0 | 25 | 0 | 22 | 2 |
| LEF1 | 0 | 0 | 25 | 0 | 19 | 4 |
| PYGO1 | 0 | 0 | 3 | 0 | 5 | 0 |

**B**

| BioID | - Wnt | | + Wnt | |
|---|---|---|---|---|
| Protein ID | WT | Ctrl | WT | Ctrl |
| BCL9 (BAIT) | 428 | 8 | 544 | 23 |
| TTF1 | 164 | 2 | 179 | 6 |
| ARID1A | 60 | 1 | 66 | 1 |
| TLE3 | 46 | 1 | 21 | 1 |
| ARID1B | 43 | 0 | 39 | 0 |
| NCOA2 | 33 | 0 | 22 | 0 |
| TLE1 | 28 | 0 | 12 | 0 |
| PYGO2 | 17 | 1 | 8 | 0 |
| LDB1 | 17 | 1 | 5 | 0 |
| PHB2 | 15 | 0 | 6 | 2 |
| NCOR1 | 13 | 1 | 10 | 1 |
| ISL2 | 10 | 0 | 9 | 0 |
| TCF4 | 7 | 0 | 18 | 2 |
| NLE1 | 6 | 0 | 4 | 0 |
| NCOA6 | 4 | 0 | 10 | 1 |
| LMX1B | 4 | 0 | 1 | 0 |
| NCOA3 | 3 | 0 | 11 | 0 |
| RBM14 | 3 | 0 | 5 | 0 |
| NCOR2 | 1 | 0 | 3 | 0 |
| APC | 0 | 0 | 29 | 0 |
| β-catenin | 0 | 0 | 18 | 2 |
| TCF3 | 0 | 0 | 14 | 0 |
| PYGO1 | 0 | 0 | 3 | 0 |
| SSBP3/4 | 0 | 0 | 1 | 0 |

**C**

| RIME | - Wnt | |
|---|---|---|
| Protein ID | WT | Ctrl |
| B9L (BAIT) | 144 | 0 |
| TLE3 | 36 | 5 |
| CBP | 7 | 0 |
| LDB1 | 3 | 0 |
| QKI | 2 | 0 |
| PYGO2 | 2 | 0 |

**D**

| BioID | - Wnt | | + Wnt | |
|---|---|---|---|---|
| Protein ID | WT | Ctrl | WT | Ctrl |
| PYGO2 (BAIT) | 375 | 0 | 381 | 0 |
| ARID1A | 217 | 1 | 249 | 0 |
| NCOR1 | 185 | 14 | 206 | 0 |
| ARID1B | 179 | 0 | 222 | 4 |
| ZNF318 | 162 | 27 | 199 | 4 |
| ARID3B | 117 | 23 | 130 | 5 |
| TLE3 | 95 | 2 | 96 | 1 |
| BCL9 | 90 | 0 | 114 | 0 |
| RBM14 | 71 | 0 | 73 | 0 |
| NCOR2 | 70 | 0 | 85 | 0 |
| SMARCA4 | 64 | 1 | 72 | 6 |
| TLE1 | 47 | 1 | 52 | 1 |
| NCOA6 | 40 | 1 | 48 | 0 |
| NCOA2 | 39 | 0 | 45 | 0 |
| LDB1 | 37 | 0 | 62 | 0 |
| SMARCA2 | 27 | 0 | 39 | 2 |
| ISL2 | 22 | 2 | 27 | 0 |
| TCF4 | 17 | 1 | 27 | 0 |
| NCOA5 | 17 | 3 | 16 | 0 |
| LMX1B | 14 | 0 | 13 | 0 |
| ARID3A | 14 | 2 | 12 | 0 |
| NCOA3 | 12 | 0 | 25 | 0 |
| LEF1 | 12 | 1 | 14 | 0 |
| GATA6 | 11 | 0 | 15 | 0 |
| SSBP3/4 | 11 | 0 | 15 | 0 |
| TCF1 | 10 | 1 | 13 | 0 |
| TCF3 | 10 | 1 | 12 | 0 |
| GATA4 | 9 | 0 | 9 | 0 |
| CBP | 4 | 0 | 13 | 1 |
| P300 | 3 | 0 | 13 | 1 |
| NCOA1 | 2 | 0 | 5 | 0 |
| ARID5B | 2 | 0 | 3 | 0 |
| UBR5 | 1 | 0 | 2 | 0 |
| SMARCD2 | 1 | 0 | 1 | 0 |
| β-catenin* | 0 | 0 | 2 | 0 |

■ Wnt enhanceosome components
■ Nuclear co-receptor components

**Figure 4.** BCL9/B9L and PYGO2 are constitutively associated with the Wnt enhanceosome, and nuclear co-receptor complexes. (**A, B**) List of BioID hits for (**A**) B9L-BirA* and (**B**) BCL9-BirA*±10–12 hr of WCM; names above the dotted line refer to the top hits, while names below this line refer to hits selected on relevance to Wnt (blue) or nuclear co-receptors (green); only specific hits with a > 5 spectral count ratio relative to the BirA* control are shown; numbers represent unweighted spectral counts (>95% probability). (**C**) RIME hits for FLAG-B9L-BirA*; only specific hits with a >5 spectral count ratio relative to the control are shown. (**D**) List of BioID hits for PYGO2-BirA*, as in (**A, B**); *, identified with lower confidence (>55% probability).

The following figure supplements are available for figure 4:

**Figure supplement 1.** Stably transfected BCL9/B9L cell lines for BioID, and summary of wt and mutant BirA* baits.

**Figure supplement 2.** Additional analysis of BioID hits.

**Figure supplement 3.** Constitutive association between B9L and TCF prior to Wnt stimulation.

cell line stably expressing B9L-BirA* at levels comparable to endogenous B9L, using FLAG resin to capture its N-terminal FLAG tag (*Figure 4—figure supplement 1*). This method relies on limited crosslinking of proteins in live cells followed by mass spectrometry of peptides cross-linked to the affinity-purified bait, and is thus capable of identifying direct ligands of the bait. It has been extensively validated for hit lists from immunoprecipitation-based approaches obtained for nuclear hormone receptors (for example, estrogen receptor) and other DNA-binding proteins (*Mohammed et al., 2016*). These RIME experiments identified only five specific hits; these proteins associated with B9L-BirA* (*Figure 4C*) include PYGO2, its only known direct ligand in the absence of Wnt. This short list also contained LDB1, which we shall identify below as another direct B9L ligand. It was topped by TLE3, suggesting that TLE3 may also be a direct ligand of B9L.

Importantly, the majority of the hits identified by B9L-BirA* or BCL9-BirA* hardly change after Wnt stimulation (*Figure 4A,B*). This implies that BCL9/B9L is associated with the Wnt enhanceosome

prior to Wnt stimulation and independently of β-catenin. We note however that the spectral counts of many hits tend to be slightly increased upon Wnt signaling, a trend that is also apparent for TLE1 and TLE3 whose labeling by B9L-BirA* was increased 2-3x upon Wnt stimulation (*Figure 4A,B*). Similarly, the labeling of GSE1 by B9L-BirA* is increased 3-5x in stimulated cells (*Figure 4A*): GSE1 is a subunit of the BRAF-HDAC complex (also known as BHC) (*Hakimi et al., 2003*), which might interact with Groucho/TLE through its HDAC subunit, potentially explaining the Wnt-dependent increase in the labeling of GSE1. In any case, TLE1/3 behaves like the LDB1 core component of the Wnt enhanceosome in remaining associated with Wnt-responsive enhancers even when these are active, consistent with recent findings that TLE1 and β-catenin can bind simultaneously to TCF1 (*Chodaparambil et al., 2014*).

## β-catenin triggers a Wnt-dependent apposition of the BCL9/B9L C-terminus to TCF

Association of β-catenin with BCL9/B9L is undetectable prior to Wnt stimulation, as expected from its low abundance in the absence of signaling. This may also explain why the labeling of CBP and p300 by B9L-BirA* is stimulated ~4–6x upon Wnt stimulation, given that these histone acetyltransferases are known ligands of β-catenin. However, both proteins exhibit a significant level of labeling even without Wnt signaling, suggesting a constitutive association with the Wnt enhanceosome, consistent with the identification of CBP by RIME as a putative direct ligand of B9L. It thus appears that the binding of CBP and p300 to β-catenin upon its docking to the Wnt enhanceosome increases their proximity to the C-terminus of BCL9/B9L, which bears the BirA* tag.

Three other factors exhibit a striking increase in labeling after Wnt stimulation, namely the TCF factors, APC and SSBP3/4 (*Figure 4A,B*). TCF4 is the only TCF paralog labeled prior to Wnt signaling, albeit at very low levels (also by the BirA* control), but its labeling by B9L-BirA* or BCL9-BirA* is Wnt-inducible by 13x or 2.5x, respectively. For TCF1/3 and LEF1, no labeling is detectable in unstimulated cells, whereas each of these factors is labeled efficiently upon Wnt signaling (>19–30x). The same is true for the APC tumor suppressor (>29x), a BCL9-specific hit that is recruited to TCF target genes upon Wnt signaling by β-catenin (*Sierra et al., 2006*), owing to direct high-affinity binding (*Choi et al., 2006*). Finally, the labeling of SSBP3/4 by B9L-BirA* is moderately increased (>6x) in Wnt-stimulated cells; as in the case of CBP/p300, SSBP3/4 is also detectable prior to Wnt signaling, albeit close to background levels. As far as we know, the expression levels of APC, SSBP3/4 and TCF factors do not change after Wnt stimulation, except for LEF1 whose levels increase ~2x in Wnt-stimulated HEK293T cells (*de la Roche et al., 2014*). Therefore, a likely explanation for the Wnt-inducible labeling of these factors is that Wnt signaling, or β-catenin, induces their proximity to the C-terminus of BCL9/B9L. We note that TCF factors are bound constitutively to their cognate enhancers (for example, TCF4 in unstimulated HEK293 cells; *Frietze et al., 2012*) although, in *Drosophila*, the association of dTCF with Wg-responsive enhancers appears to be strengthened by Wg signaling (*Parker et al., 2008*).

To obtain independent evidence for the constitutive association between BCL9/B9L and TCF, we conducted coIP assays in HEK293T cells co-expressing FLAG-B9L and GFP-TCF4. As expected, the two proteins coIP with similar efficiency in cells with or without Wnt stimulation (*Figure 4—figure supplement 3*). Likewise, coIP of GFP-BCL9 with endogenous PYGO2 is also Wnt-independent, whereas its coIP with endogenous β-catenin (or activated β-catenin, ABC) is strictly Wnt-inducible, as expected (*Figure 4—figure supplement 3*). This underscores our notion that the Wnt-inducible labeling of TCF factors by B9L-BirA* and BCL9-BirA* reflects β-catenin-dependent apposition of the C-terminus of BCL9/B9L to TCF, rather than β-catenin-dependent recruitment of Legless/BCL9 to TCF (as initially envisaged; *Kramps et al., 2002*; *Städeli and Basler, 2005*).

## The proximity of PYGO2 to TCF factors does not change during Wnt signaling

Strong independent support for this notion came from BioID experiments with PYGO-BirA*, which we expressed in T-REx-293 cells under identical conditions as the BCL9 and B9L baits; Wnt stimulation was confirmed by Western blotting against ABC (*Figure 4—figure supplement 1*). By and large, PYGO2-BirA* identified the same Wnt enhanceosome components as B9L-BirA*, with ARID1A/B topping the list (*Figure 4D*). Complete lists of specific BioID hits obtained with B9L-BirA*,

BCL9-BirA* and PYGO2-BirA* whose total number of unweighted spectral counts are ≥1, and 5x above background (that is, counts obtained for that hit with BirA*), can be found in *Supplementary files 1–3*.

SSBP3/4 was labeled more efficiently by PYGO2-BirA* than by B9L-BirA*, consistent with our evidence that Pygo binds directly to an interface shared between LDB1 and SSDP (*Fiedler et al., 2015*). Importantly, except for p300 and CBP whose labeling by PYGO2-BirA* was moderately Wnt-inducible (~3x), all other components of the Wnt enhanceosome and BAF complex were labeled with similar efficiency regardless of Wnt signaling, including TCF1/3/4 and LEF1 (*Figure 4D*). Thus, the proximity of PYGO2 to TCF factors does not change as a result of β-catenin docking the Wnt enhanceosome.

## The C-terminus of BCL9/B9L mediates binding to TLE

To identify ligands binding to HD3 or the C-terminus of BCL9/B9L, we applied BioID mass spectrometry to T-REx-293 cells stably transfected with mutant BCL9-BirA* and B9L-BirA* baits lacking these sequences (△HD3 and △C, respectively), and also with baits lacking HD1 (△HD1) expected to abrogate Pygo binding (*Fiedler et al., 2008*). We confirmed that the subcellular distributions of these mutant BCL9-BirA* and B9L-BirA* baits were not affected by the various deletions (*Figure 4—figure supplement 1*).

The lists of proteins associated with △C baits are similar to those found with the wt, but a few of them show reduced spectral counts, most notably in the case of BCL9ΔC which barely associates with TLE1 nor TLE3 (*Figure 5A*). coIP assays revealed robust binding between HA-tagged TLE3 (HA-TLE) and wt GFP-BCL9 but not with two different C-terminal truncations (*Figure 5B*). Indeed, the C-terminal WD40 domain of TLE3 is both necessary and sufficient for this coIP (*Figure 5C*). This domain binds to short motifs within a range of DNA-binding repressors (*Jennings et al., 2006*), but is not involved in binding to TCF (*Chodaparambil et al., 2014*). Therefore, BCL9/B9L could bind directly to TCF-associated TLE/Groucho (as indicated by RIME, see above).

## HD3 is a direct ligand of ChiLS

Likewise, a small number of proteins show reduced association with BCL9 or B9L baits lacking HD3, with the top hit being LDB1 whose association with B9L-BirA* is reduced by half compared to the wt control (*Figure 5D*). Indeed, binding between LDB1-FLAG and GFP-B9L, GFP-BCL9 or GFP-Lgs is readily detectable in coIP assays, but is significantly reduced if HD3 is deleted (*Figure 5E*; *Figure 5—figure supplement 1*). Binding is similarly reduced by an alanine substitution of the most conserved residue in HD3 (W472 in BCL9, W520 in B9L; see also below), and eliminated if △HD1 is tested (*Figure 5E*; *Figure 5—figure supplement 1*). This indicates the importance of HD3 and HD1 for the interaction between ChiLS and BCL9/B9L, whereby HD1 likely contributes indirectly to this interaction via its binding to PYGO2 (*Figure 5A*) (*Fiedler et al., 2008*). We generated PYGO2 KO cells by CRISPR/Cas9 (noting that HEK293T cells do not express PYGO1), to confirm that ChiLS coIPs less efficiently with BCL9 in the absence of Pygo (*Figure 5—figure supplement 2*). This supports the notion that Pygo promotes the association between BCL9/B9L and ChiLS.

Next, we asked whether HD3 is a direct binding site for ChiLS. This short conserved element is predicted to form a single α-helix (*Peng and Xu, 2011*), with an invariant tryptophan and a phenylalanine/tyrosine (F/Y) doublet further downstream whose hydrophobic side chains extend in the same direction (*Figure 6A,B*). Together, they form a hydrophobic surface to which ChiLS may bind.

To test this, we purified $^{15}$N-labelled 6xHis-Lipoyl (Lip)-tagged B9L$_{509-537}$ (Lip-HD3) after bacterial expression, for binding assays with purified ChiLS complex, as previously described (*Fiedler et al., 2015*). $^{15}$N-Lip-HD3 proved soluble and monomeric (as judged by size exclusion chromatography), and produces well-dispersed heteronuclear single-quantum correlation (HSQC) spectra. Incubation of $^{15}$N-Lip-HD3 with purified ChiLS complex, but not with purified SSDP alone, produces clear line broadening ('bleaching') of selective peaks (*Figure 6D,E*) – strong evidence for a direct interaction. Assignments of these peaks (obtained with a double-labeled $^{13}$C$^{15}$N-Lip-HD3 sample; *Figure 6—figure supplement 1*) provided experimental evidence for the helical nature of HD3 (as ascertained by TALOS+; *Shen et al., 2009*) (*Figure 6C*). It allowed us to generate a heat-map that implicates most residues of HD3 in the interaction with ChiLS (*Figure 6B*). Importantly, no spectral changes are observed if a W520R-mutant version of $^{15}$N-Lip-HD3 is incubated with ChiLS (*Figure 6F*), showing

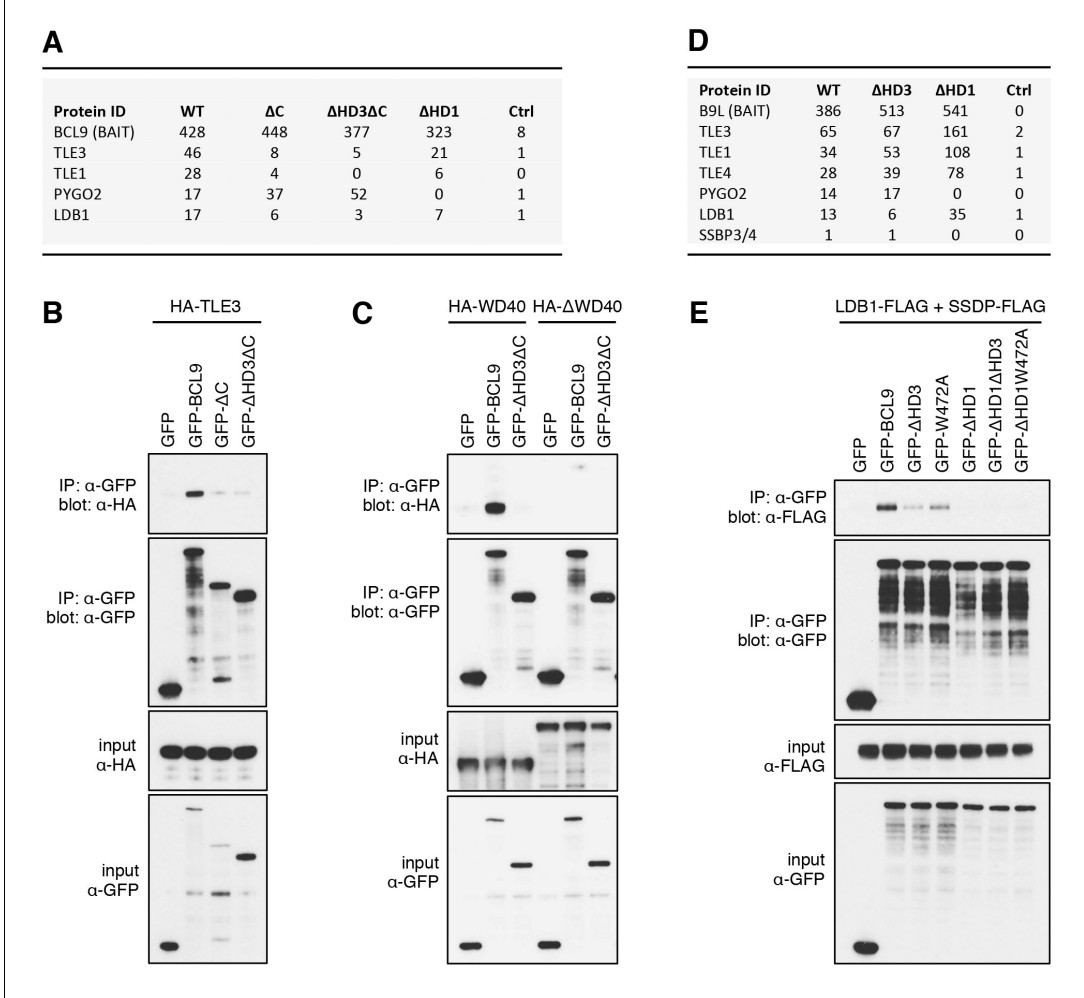

**Figure 5.** The Legless/BCL9 C-terminus binds to core Wnt enhanceosome components. (**A**) Top BioID hits showing differential association with wt versus mutant BCL9-BirA* (unweighted spectral counts > 95% probability). (**B, C**) Western blots of coIPs of wt or mutant GFP-BCL9 with (**B**) HA-TLE3 or (**C**) HA-tagged truncations, after co-expression in HEK293T cells (lysed 48 hr after transfection), probed with antibodies as indicated on the left. (**D**) Top differential BioID hits of B9L-BirA* as in (**A**). (**E**) CoIP assays between co-expressed wt or mutant GFP-BCL9, LDB1-FLAG and SSDP-FLAG, as in (**B**); the band in the top panel corresponds to LDB1-FLAG.

The following figure supplements are available for figure 5:

**Figure supplement 1.** HD3-dependent interaction between LDB1 and Legless/B9L.

**Figure supplement 2.** CRISPR/Cas9-based gene editing strategy for *PYGO2* in HEK293T cells.

that this point mutation abrogates the binding of HD3 to ChiLS. Thus, ChiLS binds directly and specifically to HD3.

Given this physical interaction between HD3 and ChiLS, we asked whether we could also detect genetic interactions between *lgs* and *chip* in *Drosophila*. Flies heterozygous for *chip* exhibit multiple wing notches (*Shoresh et al., 1998*) that are strongly exacerbated by heterozygosity of *ssdp*; however, this phenotype is ameliorated by heterozygosity of several Wnt signaling components including *pygo*, *groucho* and *lgs* (*Figure 6—figure supplement 2*). Interestingly, the strongest interaction is seen with *lgs^ΔHD3* whose heterozygosity restores normal margins in >25% of the flies. The same is also seen with *lgs^2-8* heterozygosity although full suppression is less penetrant (*Figure 6—figure supplement 2*). These strong genetic interactions between *chip* and HD3-defective *lgs* alleles further underscore the close functional link between these two proteins.

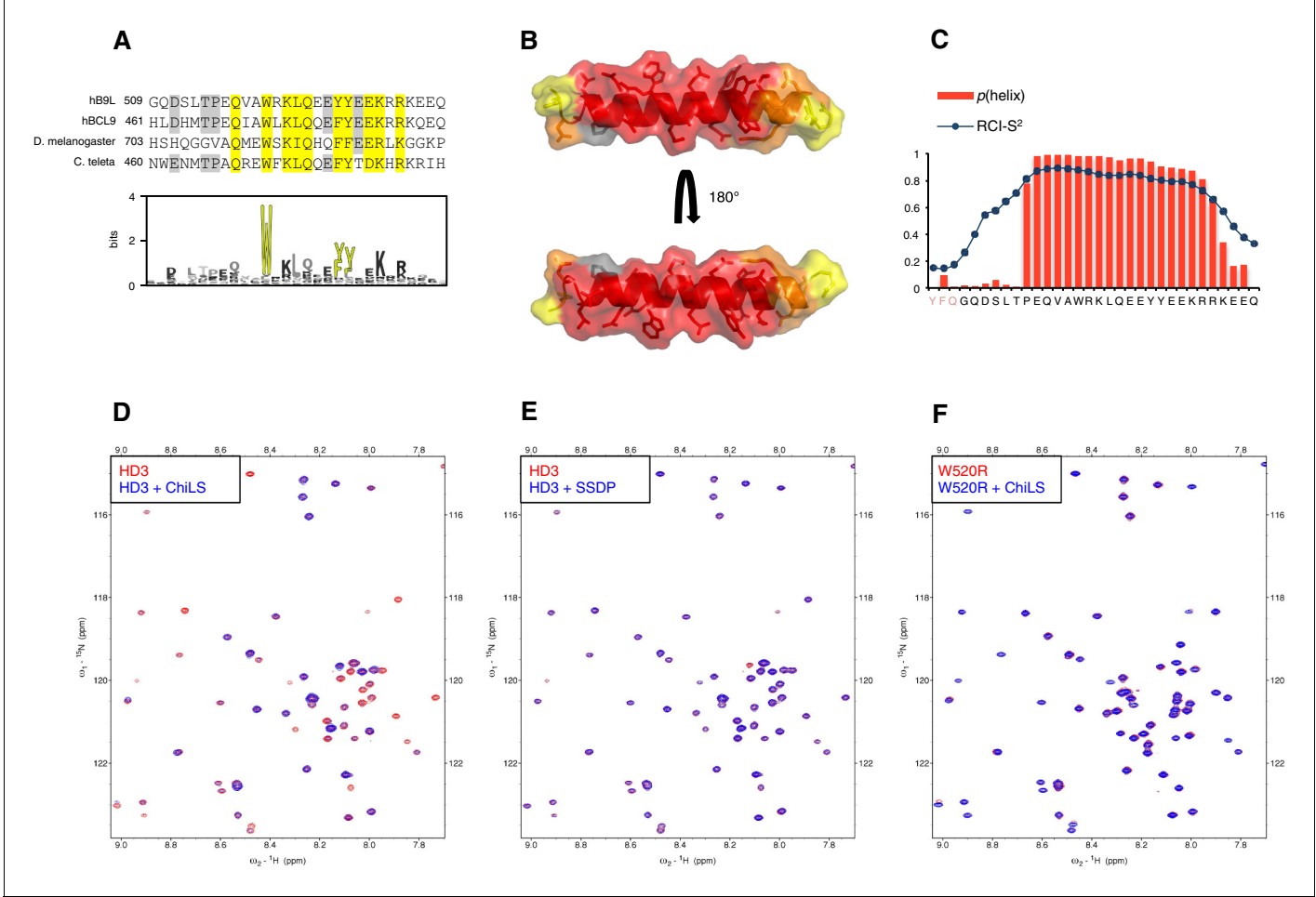

**Figure 6.** HD3 binds directly to ChiLS. (**A**) *Top*, sequence alignments of HD3; *yellow*, conserved residues; *grey*, semi-conserved residues. *Bottom*, position-specific alignment (HMMER; *Finn et al., 2011*) for B9L HD3; *yellow*, conserved tryptophan and phenylalanine/tyrosine doublet. (**B**) Predicted structure of HD3 (by I-TASSER), with heat-map indicating relative line broadenings upon incubation of $^{15}$N-HD3 with ChiLS (see **D**), ranging from 80% (*red*) to 40% (*yellow*); *grey*, proline (not detectable). (**C**) TALOS+ predictions of α-helicity of HD3, based on backbone secondary chemical shifts in *Figure 6—figure supplement 1*; for each position, the rigidity index (RCI-S$^2$) and helical probability (**p**) are indicated (*pink*, N-terminal linker residues). (**D–F**) Overlays of HSQC spectra of 100 μM $^{15}$N-labeled (**D, E**) wt HD3 or (**F**) W520R mutant alone (*red*), and probed with 300 μM (**D, F**) MBP-Chip$_{205-436}$-Lip-SSDP$_{1-92}$ or (**E**) Lip-SSDP$_{1-92}$ (*blue*); line broadening owing to ChiLS binding to wt HD3 (**D**) results in the disappearance of selected resonances in the blue (HD3 + ChiLS) spectrum, revealing the corresponding resonances from the red (HD3-only) spectrum.

The following figure supplements are available for figure 6:

**Figure supplement 1.** Assignment of the [$^1$H-$^{15}$N]-HSQC spectrum of HD3.

**Figure supplement 2.** Modification of the *chip* phenotype by loss of HD3.

## The Wnt enhanceosome associates with nuclear co-receptor complexes

Unexpectedly, we consistently found multiple components of nuclear co-receptors complexes amongst the top hits for all three BirA* baits, including several NCOA co-activators and NCOR co-repressors (*Figure 4*). In the case of B9L-BirA*, we also found the DNA-binding components of the resident complex, namely the arylhydrocarbon receptor (AHR) and its partner ARNT (*Figure 4A*). Clearly, there is close proximity between BCL9/B9L-PYGO2 and nuclear co-receptor complexes. This suggests that a substantial fraction of Wnt-responsive enhancers also contain binding sites for nuclear receptors (for example, AHR in HEK293T cells), and that these sites are near TCF-binding sites.

## Discussion

Our study has uncovered genetic and physical interactions between two constitutive core components of the Wnt enhanceosome and the C-terminus of Legless/BCL9. The first of these is ChiLS, the core module of the Wnt enhanceosome (*Fiedler et al., 2015*) (*Figure 7*): we have shown that ChiLS is a direct and specific ligand of the α-helical HD3 element of B9L and, likely, of other Legless/BCL9 orthologs, given the strong sequence conservation of this α-helix (*Figure 6*). The physiological relevance of this interaction with ChiLS is underscored by genetic analysis in flies. Our evidence thus implicates HD3 as an evolutionary conserved contact point between Legless/BCL9 and ChiLS, although the primary link between these two proteins appears to be provided by Pygo.

A second link between the Legless/BCL9 C-terminus and the Wnt enhanceosome is mediated by the WD40 domain of TLE/Groucho. Given our evidence from RIME, this link is also likely to be direct although, for technical reasons, we have not been able to prove this. The function of the C-terminus of Legless/BCL9 for transducing Wnt signals was revealed by the *wg*-like phenotypes in *Drosophila* larvae and flies and by their defective transcriptional Wg responses (*Figure 1 and 2*), and by the loss of transcriptional Wnt responses in BCL9/B9L-deleted human cells (*Figure 3*). Our evidence indicates that Legless/BCL9 undergoes three separate functionally relevant interactions with distinct components of the Wnt enhanceosome—with Pygo, ChiLS and Groucho/TLE (*Figure 7*). Importantly, BioID revealed that these interactions are constitutive, preceding Wnt signaling, and that they hardly change upon Wnt stimulation (*Figure 4*). Taken together with its multivalent interactions with the Wnt enhanceosome, this is consistent with Legless/BCL9 being a core component of this complex, providing a scaffolding function that facilitates its assembly and/or maintains its cohesion.

Following Wnt stimulation, Legless/BCL9 undergoes an additional physiologically relevant interaction, by binding to (stabilized) Armadillo/β-catenin via HD2 (*Kramps et al., 2002*). Legless/BCL9 thus confers Wnt-responsiveness on the Wnt enhanceosome through its ability to capture Armadillo/β-catenin. In other words, in addition to scaffolding the enhanceosome, Legless/BCL9 also earmarks this complex for Wnt responses. Intriguingly, our BioID data indicated that the capture of β-catenin by Legless/BCL9 triggers its rearrangement within the complex, apposing its C-terminus to TCF (*Figure 7*). This apparent β-catenin-dependent apposition is consistent with structural data showing that BCL9/B9L HD2 is closely apposed to TCF when in a ternary complex with β-catenin (*Sampietro et al., 2006*). Our evidence support the notion of Legless/BCL9 acting as an 'Armadillo loading factor', facilitating access of Armadillo/β-catenin to TCF (*de la Roche and Bienz, 2007*; *Townsley et al., 2004*), but argues against the original co-activator hypothesis which posited that Legless/BCL9 is recruited to TCF by Armadillo/β-catenin exclusively in Wnt-stimulated cells (*Kramps et al., 2002*; *Städeli and Basler, 2005*). Whatever the case, the β-catenin-dependent apposition of the Legless/BCL9 C-terminus to TCF is likely to trigger Wnt enhanceosome switching from OFF to ON, resulting in the relief of Groucho/TLE-dependent repression and culminating in the Wnt-dependent transcriptional activation of linked target genes (*Figure 7*).

This transition of the Wnt enhanceosome from OFF to ON is accompanied by a proximity gain between Legless/BCL9 and CBP/p300 (*Figure 4*), likely to reflect at least in part its de novo binding to Armadillo/β-catenin. However, our evidence indicates that CBP/p300 is associated with the Wnt enhanceosome prior to Wnt signaling, possibly via direct binding to B9L as suggested by RIME (*Figure 4C*), and that the docking of Armadillo/β-catenin to the Wnt enhanceosome strengthens its association with CBP/p300, and/or directs the histone acetyltransferase activity of CBP/p300 towards its substrates, primarily the histone tails. By acetylating these tails, CBP/p300 appears to promote Wnt-dependent transcription in flies and human cells (*Li et al., 2007*). Indeed, CBP-dependent histone acetylation has been observed at Wg target enhancers in *Drosophila* although, interestingly, this preceded transcriptional activation (*Parker et al., 2008*). This is consistent with our BioID data, indicating constitutive association of CBP/p300 with the Wnt enhanceosome.

It seems plausible that histone acetylation at Wnt target enhancers is instrumental in antagonizing the compaction of their chromatin imposed by Groucho/TLE, which depends on its tetramerization via its Q domain (*Chodaparambil et al., 2014*) as well as its binding to HDACs (*Jennings et al., 2008*; *Turki-Judeh and Courey, 2012*). Indeed, we found HDACs near the bottom of our BioID lists, and one of the top hits identified by B9L was GSE1, a subunit of the BRAF-HDAC complex (*Hakimi et al., 2003*). However, CBP/p300 also has non-histone substrates within the Wnt enhanceosome, including dTCF in *Drosophila* whose Armadillo-binding site can be acetylated by dCBP, which

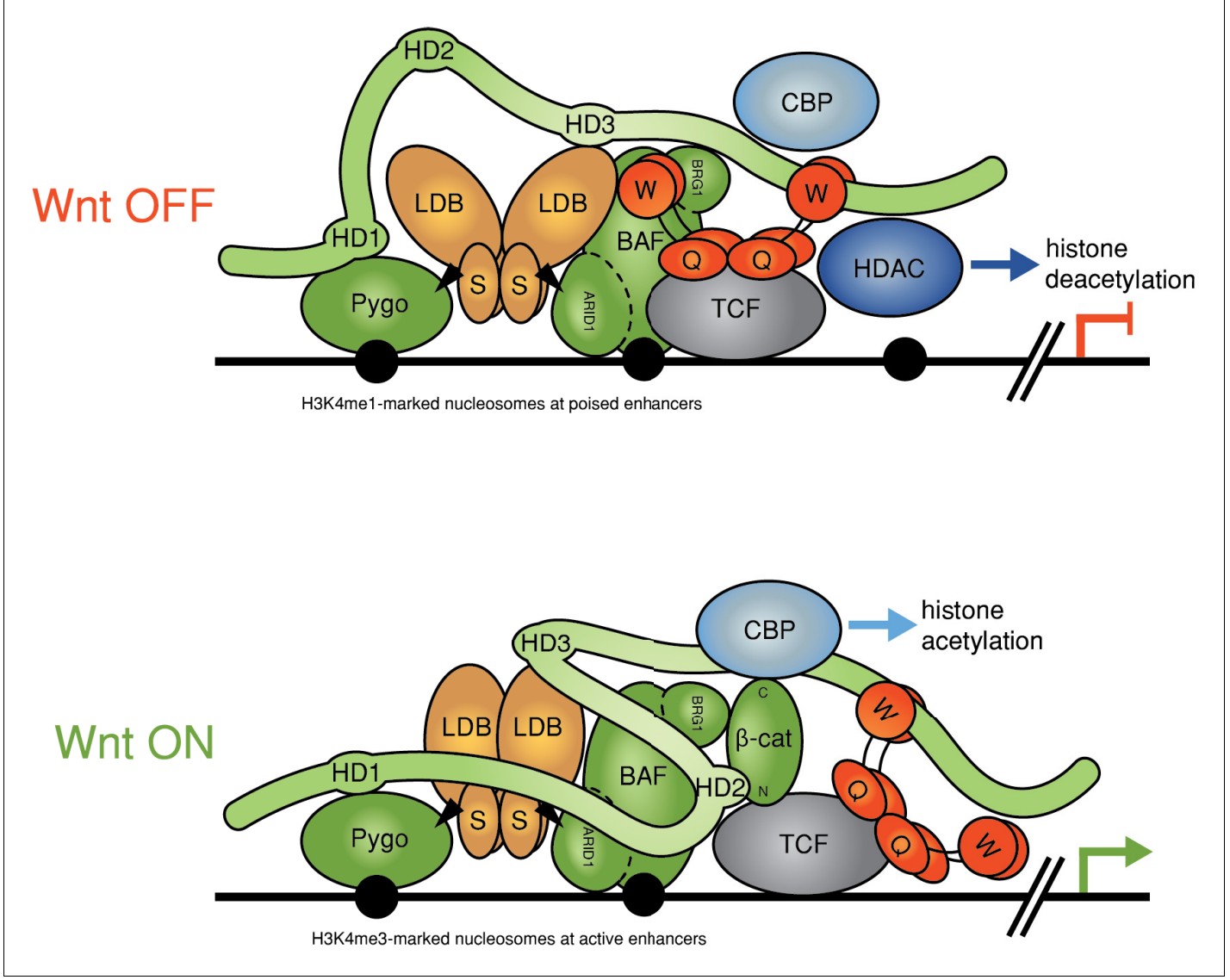

**Figure 7.** Refined model of the Wnt enhanceosome. The Wnt enhanceosome complex associated with a Wnt-responsive enhancer in its OFF or ON state (for initial model, see *Fiedler et al. [2015]*, illustrating multivalent constitutive interactions of Legless/BCL9 with Pygo (through HD1), ChiLS (through HD3) and Groucho/TLE (through its C-terminus). *OFF*, HD2 is poised to interact with Armadillo/β-catenin upon Wnt-induced stabilization; *ON*, rearrangement of Legless/BCL9 upon recruitment of Armadillo/β-catenin, resulting in the apposition of its C-terminus to TCF. CBP/p300 is associated with both states (possibly through direct binding to Legless/BCL9); its activity may be directed towards histones upon Armadillo/β-catenin binding, which antagonizes Groucho/TLE-dependent silencing and promotes the transcription of linked target genes. Likewise, the BAF complex is associated with both states (through the NPF motif of its subunit Osa/ARID1), earmarking the complex for feedback inhibition (see *Figure 7—figure supplement 1*). S, SSDP; Q, Q domain of TLE (tetramerizing, and binding to TCF); W, WD40 domain of TLE (binding to ChiLS); arrows, NPF-mediated interactions; green, positively-acting components; red, negatively-acting components; black circles, nucleosomes bearing H3K4me marks of poised or active enhancers (*Kharchenko et al., 2011*).

The following figure supplement is available for figure 7:

**Figure supplement 1.** Reinstalling silencing on a Wnt-responsive enhancer.

thus blocks the binding between the two proteins (*Waltzer and Bienz, 1998*) and antagonizes Wg responses (*Li et al., 2007*). It thus regulates Wnt-dependent transcription positively as well as negatively, similarly to Groucho/TLE which not only silences Wnt target genes but also earmarks them for Wnt inducibility, as a core component of the Wnt enhanceosome. It is intriguing that both bimodal

regulators are associated constitutively with this complex. A corollary is that the docking of Armadillo/β-catenin to the Wnt enhanceosome changes their substrate specificities and/or activities.

An important refinement of our initial enhanceosome model is with regard to the BAF complex, which appears to be a constitutive component of the Wnt enhanceosome (*Figure 7*), as indicated by our BioID data. This complex is highly conserved from yeast to humans, and it contains 15 subunits in human cells (*Kadoch and Crabtree, 2015*), including the DNA-binding Osa/ARID1 subunit. A wealth of evidence from studies in flies and mammals indicates that this complex primarily antagonizes Polycomb-mediated silencing of genes, most notably of the *INK4A* locus which encodes an anti-proliferative factor, which could explain why the BAF complex functions as a tumor suppressor in many tissues. However, recall that this complex also specifically antagonizes Armadillo/β-catenin-mediated transcription (*Collins and Treisman, 2000*), likely via its BRG/BRM subunit which directly binds to β-catenin (*Barker et al., 2001*). Evidence from studies in *Drosophila* of Wg-responsive enhancers suggests that this complex mediates a negative feedback from high Wg signaling levels near Wg-producing cells which results in re-repression (*Collins and Treisman, 2000*), imposed by the Brinker homeodomain repressor (*Saller et al., 2002*; *Waltzer et al., 2001*; *Theisen et al., 2007*) and its Armadillo-binding Teashirt co-repressor (*Gallet et al., 1999*) (*Figure 7—figure supplement 1*). The same factors may also instal silencing on Wnt-responsive enhancers upon cessation of Wnt signaling. Notably, mammals do not have a Brinker ortholog, which could explain some of the apparent functional differences between flies and mammals with regard to the BAF complex (*Kadoch and Crabtree, 2015*). However, the closest mammalian relatives of Teashirt are the Homothorax/MEIS proteins, a family of homeodomain proteins whose expression can be Wnt-inducible (for example, *Wernet et al., 2014*). They are thus candidates for Wnt-induced repressors that confer BAF-dependent feedback inhibition.

Notably, none of our BioID lists contained RUNX proteins. Based on our functional evidence from *Drosophila* midgut enhancers, we proposed that these proteins (which bind to both enhancers and Groucho/TLE) are pivotal for initial assembly of the Wnt enhanceosome at Wnt-responsive enhancers during early embryonic development, or in uncommitted progenitor cells of specific cell lineages (*Fiedler et al., 2015*). However, HEK293 cells are epithelial cells and may thus not express any RUNX factors. In any case, our negative BioID results suggest that RUNX factors function in a hit-and-run fashion. Evidently, the Wnt enhanceosome complex, once assembled at Wnt-responsive enhancers, can switch between ON and OFF states without RUNX.

In summary, we have uncovered a fundamental role to Legless/BCL9 as a scaffold of the Wnt enhanceosome, far beyond its role in linking Armadillo/β-catenin to Pygo. Indeed, the function of Legless/BCL9 may extend beyond transcriptional Wnt responses, as indicated by the unexpected discovery of its strong association with nuclear co-receptor complexes. Potentially, these associations underlie the observed cross-talk between Wnt/β-catenin and nuclear hormone receptor signaling, documented extensively in the literature (for example, *Beildeck et al., 2010*; *Mulholland et al., 2005*; *Schneider et al., 2014*), including evidence for direct activation of the androgen receptor by β-catenin (*Kypta and Waxman, 2012*). Furthermore, a strong association between TLE1 and the estrogen receptor has been discovered in breast cancer cells, where TLE1 assists the estrogen receptor in its interaction with chromatin and its proliferation-promoting function (*Holmes et al., 2012*). This is reminiscent of the role of Groucho/TLE as a cornerstone of the Wnt enhanceosome, proposed to earmark TCF enhancers for subsequent β-catenin docking and transcriptional Wnt responses (*Fiedler et al., 2015*). It will be interesting to test experimentally the putative roles of BCL9/B9L and Pygo in enabling cross-talk between β-catenin and nuclear hormone receptor signaling, both during normal development and in cancer.

## Materials and methods

### Plasmids, antibodies and resins

The following plasmids have been described: LDB1-FLAG, SSDP-FLAG (*Fiedler et al., 2015*); murine FLAG-B9L, FLAG-ΔC (called FLAG-ΔCter), GFP-B9L (*Adachi et al., 2004*); GFP-Lgs, GFP-TCF4 (*Townsley et al., 2004*). Human GFP-BCL9 was generated by exchanging the FLAG tag of FLAG-BCL9 (*de la Roche et al., 2008*). Mutants were made from parental plasmids using standard site-directed mutagenesis procedures, and confirmed by sequencing. TLE3 was amplified by PCR from

6xMyc-TLE3 (*Hanson et al., 2012*) and subcloned in pcDNA3.1-N-HA, and mutants (WD40, *Δ*WD40) were generated from this subclone.

The following antibodies and resins were used: α-GFP RRID:AB_439690, α-FLAG RRID:AB_262044, α-HA RRID:AB_390918, α-*β*-actin (for human lysates) RRID:AB_476744 (Sigma-Aldrich, St. Louis MO, USA); α-BCL9 RRID:AB_2063609, α-BCL9-2 RRID:AB_2063747 (Abingdon, UK); α-*β*-actin (for fly lysates) RRID:AB_2305186, α-BirA RRID:AB_300830, α-Pygo2 RRID:AB_10863482 (Abcam, Cambridge, UK); α-ABC RRID:AB_11127203 (Cell Signaling Technology). Rat α-Lgs antiserum (Eurogentec, Liège, Belgium) was obtained from pre-bleed immunizations with gluthathione S-transferase-purified Lgs$_{232-555}$. GFP-Trap resin RRID:AB_2631357 (Chromotek, Planegg, Germany) was used for coIP assays, and α-FLAG M2 affinity gel RRID:AB_10063035 (Sigma-Aldrich) was used for RIME.

## CRISPR/Cas9 genome editing in flies

The CRISPR design tool at *crispr.mit.edu* was used to design single-stranded oligomers for sgRNA targeting vectors. pCFD3-1S and pCFD3-2S were generated by hybridizing single-stranded oligomers with their complementary strands in 2 mM Tris-HCl (pH 7.4), 10 mM NaCl, 200 μM EDTA at 95°C for 5 min, and by subsequently ligating the double-stranded oligomers into a *BbsI* restriction site of pCFD3 (kindly provided by Simon Bullock, MRC LMB, Cambridge, UK). pCFD4-HD3 was generated by inserting a double-stranded oligomer coding for sgRNAs targeting upstream and downstream of HD3 (*Figure 1—figure supplement 1*) into pCFD4 (from Simon Bullock) using Gibson assembly (*Gibson et al., 2009*).

Prior to injections into fly embryos (from a *vermilion* strain), pCFD3-1S, pCFD3-2S and pCFD4-HD3 were purified using plasmid midi kit columns (Qiagen, Venlo, Netherlands), and dissolved in injection buffer at 200 ng μl$^{-1}$ containing 100 μM phosphate buffer and 5 mM KCl. Dechorionated embryos were injected by standard procedures, and sgRNA-expressing transgenic lines were identified on the basis of their *vermilion*$^{+}$ eye color, and subsequently crossed to a transgenic fly strain bearing *act5c.cas9* (from Simon Bullock), essentially as described (*Port et al., 2014*, *2015*). For genotyping, DNA was extracted from individual flies by standard methods, and *lgs* sequences were determined after PCR amplification using the following primers: 5'-CATCGGGAAGAACAGTTGGC-3' and 5'-GGACTGGATGCAGCAAATCG-3' (for PCR amplification), and 5'-TGAATCAATTTC TTTTTCCTG-3' (for sequencing; see also below).

## CRISPR/Cas9 genome editing in HEK293T cells

HEK293T cells were purchased from the European Collection of Cell Cultures (authenticated by STR DNA profiling). Upon receipt, cells were frozen, and individual aliquots were taken into culture, typically for analysis within <10 passages. Single KO (of *BCL9*, *B9L*, or *PYGO2*) and *BCL9/B9L* DKO cells were generated essentially as described (*Ran et al., 2013*), using guide RNA-encoding plasmid derivatives of pSpCas9(BB)−2A-GFP (PX458) obtained as described above for their fly equivalents (*Figure 3—figure supplement 1*; *Figure 5—figure supplement 2*). Cells were sorted 48 hr post-transfection at a density of 1 cell/well in a 96-well plate and grown in Dulbecco's modified Eagles medium (supplemented with 10% fetal bovine serum, 100 U ml$^{-1}$ penicillin, 100 μg ml$^{-1}$ streptomycin) for 20–25 days to isolate individual clones. These were screened by genotyping (see below) and Western Blot analysis (for lack of BCL9, B9L or PYGO2 expression).

For genotyping, 2–5 × 10$^4$ HEK293T cells were homogenized in 15 μl MicroLYSIS-Plus (Microzone, Haywards Heath, UK) and thermocycled as specified by the manufacturers. 2 μl supernatant was used for PCR amplification, and the resulting PCR products were purified using the QIAquick purification kit (Qiagen, Hilden, Germany) and sequenced. Sequence chromatograms were analyzed using MacVector software (MacVector Inc., Cary NC, USA). The following primers were used: 5'-CGAGATTTTCCTCTGGCAGC-3' and 5'-AAGGAGTCGGCGGAAATACT-3' (amplification of *BCL9*); 5'-GGATCCTGGCTAACAAGACAAG-3' and 5'-AGAAGTCCGACCACTCTGTG-3' (amplification of *B9L*); 5'-AGTCCAGAAAAGAAGCGAAGG-3' and 5'-CAGAAGCTTCAGTGGTCAGC-3' (amplification of *PYGO2*); 5'-ACCCACTTCCACAGCAGAG-3' (sequencing of *BCL9*); 5'- TGTCTGAGGAAGCCA TGGAG-3' (sequencing of *B9L*); 5'-CTCGATCTCCTGACCTCGTG-3' (sequencing of *PYGO2*).

## Fly strains and analysis

The following mutant *Drosophila* strains were used: *lgs*[20F](**Kramps et al., 2002**); *chip*[e55] (**Morcillo et al., 1997**); *ssdp*[L7] (**van Meyel et al., 2003**); *dTcf*[3] (**van de Wetering et al., 1997**); *arm*[XM19] (**Peifer and Wieschaus, 1990**); *gro*[MB5] (**Jennings et al., 2008**); *pygo*[S123] (**Thompson et al., 2002**). The strains for CRISPR-mediated genome engineering (*act5c.cas9*, *vermilion* bearing an *attP2* landing site) were provided by Simon Bullock (see also **Port et al. [2015]**). Larvae were raised at 22°C for scoring of pupation, survival and mutant phenotypes; homozygous flies (**Figure 1**) were generated from homozygous mothers where possible (to eliminate maternal contributions). Preparation of wings, legs and abdomens were done according to standard protocols, and bright-field images were acquired with a Zeiss Axiophot microscope.

Wing and leg discs dissected from late third instar larvae (or pupating larvae if specified) were fixed with paraformaldehyde, and stained with α-Sens (**Nolo et al., 2000**), α-Wg RRID:AB_528512 (Developmental Studies Hybridoma Bank) or chicken α-β-galactosidase RRID:AB_2313752 (Immune Systems, Paignton, UK) as described (**Fiedler et al., 2015**; **Miller et al., 2013**). All discs were counter-stained with DAPI, to control for the focal plane (though DAPI images are not shown in triple antibody stainings), and single confocal images were acquired at identical settings with a Zeiss Confocal Microscope. RNA was isolated from dissected wing discs with the RNeasy kit (Qiagen, Hilden, Germany) and converted to cDNA with Super RT (HTBiotechnology Ltd, Cambridge, UK). RT-qPCR reactions were subsequently run in Fast-96-well format on a Vii7 Real-Time PCR System (Applied Biosystems, Foster City CA, USA). The following TaqMan probes were used: Dm01792952_m1, Dm01820389_m1, Dm01804677_m1, Dm01842959_m1, Dm01803388_m1, Dm01843776_s1 and Dm02151827_g1 (Applied Biosystems, Foster City CA, USA). Statistical significance was assessed with two-tailed Student's t-tests.

## Mass spectrometry

To generate BioID plasmids, BirA*(R118G) was amplified from pcDNA3.1 mycBioID (**Roux et al., 2012**) by PCR and subcloned into pcDNA5/FRT/TO using megaprimer PCR. Coding sequences for PYGO2, BCL9 and B9L (and mutants thereof) were amplified likewise and inserted directly upstream of BirA* in pcDNA5/FRT/TO using Gibson assembly (**Gibson et al., 2009**). For each stably transfected BioID cell line, $1.4–2.1 \times 10^8$ cells were grown adherent to full confluence, washed once with phosphate-buffered saline (PBS), flash-frozen in liquid nitrogen, and stored at −80°C for 1–20 days (for further details, see **Al-Jassar et al., 2017**). BioID pull-downs were then essentially done as described (**Roux et al., 2013**), and protein was eluted from the beads by boiling for 15 min in LDS sample buffer (ThermoScientific, San Jose CA, USA). RIME pull-downs were essentially done as described (**Mohammed et al., 2016**), except that $3–5 \times 10^7$ cells were washed in PBS and incubated for 6 min in 4% methanol-free formaldehyde (Polysciences, Warrington, USA) before lysis and boiling in sample buffer. All samples were resolved on 4–12% Bis-Tris polyacrylamide gels (Life Technologies, Carlsbad CA, USA), and gels were stained with Imperial Protein Stain (ThermoScientific, San Jose CA, USA). Gel slices (2–3 mm) were prepared for mass spectrometric analysis by manual in situ digestion with trypsin, and digests were analyzed by nano-scale capillary LC-MS/MS using an Ultimate U3000 HPLC (ThermoScientific Dionex, San Jose, USA). The analytical column outlet was directly interfaced via a nano-flow electrospray ionization source, with a hybrid dual pressure linear ion trap mass spectrometer (Orbitrap Velos, ThermoScientific, San Jose, USA). LC-MS/MS data were searched against a protein database (UniProt KB) using the Mascot search engine program (Matrix Science, London, UK). MS/MS data were validated using the Scaffold program (RRID:SCR_014345; Proteome Software Inc., Portland OR, USA), and processed with R (RRID:SCR_001905).

## Cell-based assays

HEK293T cells were cultured and transfected essentially as described (**Metcalfe et al., 2010**). For coIP assays, cells were lysed 48 hr post-transfection in lysis buffer (20 mM Tris–HCl pH 7.4, 10% v/v glycerol, 100 mM NaCl, 1 mM EDTA, 5 mM NaF, 2 mM $Na_3VO_4$, 0.2% v/v Triton-X-100) supplemented with protease inhibitors (Roche, Basel, Switzerland), and sonicated twice for 10 s with an amplitude of 15 μm using a Soniprep 150 plus (MSE, London, UK). Samples were cleared by centrifugation at 16,100 x g for 10 min, and supernatants were incubated with resin for 2 hr at 4°C. All inputs shown are equivalent to 10% of IPs.

For luciferase reporter assays, SuperTOP (*Veeman et al., 2003*) was co-transfected with CMV-Renilla as internal control, and assays were performed initially 48 hr post-transfection, but subsequently 24 hr post-transfection (for all figures shown in this study). Wnt inductions were for 6 hr (unless specified otherwise), either with Wnt3a-conditioned-media (WCM) or 20 mM LiCl (or 20 mM NaCl as control), and values of uninduced cells were set to 1.

RNA extractions, cDNA synthesis and RT-qPCR reactions were conducted as described above for fly wing discs. The following TaqMan probes were used: Hs00610344_m1, Hs01370227_mH and Hs00427620_m1 (Applied Biosystems, Foster City CA, USA).

## Protein purification and NMR spectroscopy

The expression and purification of 6xHis-MBP-LDB1$_{56-285}$, 6xHis-Lip-SSDP$_{1-92}$ and 6xHis-Lip-HD3$_{509-537}$ (from human B9L) as well as the acquisition and analysis of [$^1$H-$^{15}$N]fast-HSQC spectra were done as described (*Fiedler et al., 2015*, *2008*; *Miller et al., 2013*). Spectra were acquired on a Bruker Avance-3 spectrometer operating at 600 MHz $^1$H frequency and with a sample temperature of 298 °K. All samples were prepared in aqueous phosphate buffer of physiological ionic strength (25 mM phosphate pH 6.7, 150 mM sodium chloride). Backbone resonance assignments were obtained for [$^{13}$C-$^{15}$N]-double-labelled Lip-HD3 using standard procedures (*Fiedler et al., 2015*, *2008*; *Miller et al., 2013*).

## Acknowledgements

We thank Mark Skehel and his team for the mass spectrometry; Maria Daly and her team for cell sorting; Simon Bullock and Philip Port for fly CRISPR reagents and advice; Mark van Breugel, Marc de la Roche and Jason Carroll for technical advice and discussion. This work was supported by the Medical Research Council (U105192713) and by Cancer Research UK (C7379/A8709 and C7379/A15291).

## Additional information

### Funding

| Funder | Grant reference number | Author |
| --- | --- | --- |
| Medical Research Council | U105192713 | Mariann Bienz |
| Cancer Research UK | C7379/A8709 | Mariann Bienz |
| Cancer Research UK | C7379/A15291 | Mariann Bienz |

The funders had no role in study design, data collection and interpretation, or the decision to submit the work for publication.

### Author contributions

LMvT, JM, Conceptualization, Resources, Data curation, Formal analysis, Validation, Investigation, Visualization, Methodology, Writing—review and editing; MF, TJR, Resources, Formal analysis, Validation, Investigation, Methodology; MB, Conceptualization, Formal analysis, Supervision, Funding acquisition, Validation, Writing—original draft, Project administration, Writing—review and editing

### Author ORCIDs

Mariann Bienz, http://orcid.org/0000-0002-7170-8706

## Additional files

### Supplementary files

• Supplementary file 1. Complete list of specific BioID hits obtained with B9L-BirA*.

• Supplementary file 2. Complete list of specific BioID hits obtained with BCL9-BirA*.

• Supplementary file 3. Complete list of specific BioID hits obtained with PYGO2-BirA*.

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
