## [Decision Letter]

Thank you for submitting your article "Scaffolding of multiple Wnt enhanceosome components by BCL9/Legless" for consideration by *eLife*. Your article has been reviewed by two peer reviewers, and the evaluation has been overseen by Robb Krumlauf as Reviewing Editor and Fiona Watt as the Senior Editor. The reviewers have opted to remain anonymous.

The reviewers have discussed the reviews with one another and the Reviewing Editor has drafted this decision to help you prepare a revised submission.

Summary:

This paper presents interesting data that has the potential to help to resolve the mechanism of transcriptional activation of Wnt target genes. The manuscript addresses an important mechanistic detail of the TCF/b-catenin complex, concerning the addition of the ChiLS complex and how the Wnt "enhanceosome" may be assembled. The dominant model on TCF/b-catenin mediated transcription is the "chain of co-activator" model, which posits that TCF recruits β-catenin that in turn recruits the BCL9 and Pygo complex and other co-activators. The authors have challenged this model through a series studies, including the *eLife* 2015 paper, and they have argued that the BCL9, Pygo and the new ChiLS complex is preassembled on Wnt enhanceosome to recruit β-catenin. Understanding the mechanism concerning TCF/β-catenin is of general interest and important so the paper in principle is worthy of publication in *eLife*. However, there are a number of aspects of the work which need to be strengthened to make a compelling case. A few additional experiments should allow the authors to provide the supporting evidence to solidify this model and the authors need to modify the text and discussion around their model where the data are inconsistent. We realize it may take time to conduct the two main experiments suggested but we are willing to allow that time to strengthen this story.

The specific points to be addressed are:

1) The authors do not discuss some inconsistencies of their data with current models of Wnt signaling. They need to provide a more comprehensive interpretation of their data and how it might be reconciled with their previous studies. If the C-terminal region of Bcl9 proteins functions by binding the repressor TLE, one would expect deleting this region to enhance basal expression of Wnt target genes, rather than reducing their activation by Wnt. Analysis of Wg target gene expression in the *lgs^2-8^* mutant might help to clarify this issue.

It is also not clear why the Chip heterozygous phenotype, which looks like a loss of wg function, should be suppressed by removing positively acting components of the complex such as *arm* and *lgs*, as well as the negatively acting component gro.

The absence of any Runx protein in the BCL9 and B9L biotinylation assays argues against such a protein being a central part of the Wnt enhanceosome as proposed by Fiedler et al.

The observation that Wnt signaling is necessary for the interaction of Lgs with β-catenin and TCF proteins and does not reduce its association with TLE proteins does not support a model in which the enhanceosome is present at TCF-bound enhancers in the absence of Wnt signaling.

2) The Authors employed BioID (using BirA*) followed by mass spectrometry to identify BCL9 (and the related B9L) binding partners (TLE, ChiLS, and nuclear receptor co-activators). They went one step further and combined BioID and spectral counting as a (semi) quantitative measurement for determination of protein interactions with and without Wnt. It is not clear to me whether these methods offer reliable quantification. For example, spectral counts for LDB1 and SSDP in Figure 3 appear to contradict the IP results in Figure 3, in particular with regard to the HD1 deletion mutant. It would be highly desirable that additional approaches can back up these arguments (derived from BioID). One of the key arguments of the authors' model is that Pygo/BCL9 and associated ChiLS are preassembled on chromatin prior to Wnt stimulation and recruits b-catenin to TCF upon Wnt signaling. The authors should demonstrate this important issue using other techniques, such as CHIP.

3) The NMR results showed that HD3 directly interacts with LDB1/SSDP, and HD3 deletion and W472A mutants reduced BCL9-LDB1 coIP. But HD1 appeared to have a more critical role as its deletion completely abolished BCL9-LDB1 coIP. The authors suggested, reasonably, that HD1 binding to Pygo, which interacts with LDB1, has a critical role in this multivalent interaction. The authors should be able to demonstrate this point directly using Pygo knockdown or knockout (via CRISPR).

---

## [Author Response]

*Summary:*

This paper presents interesting data that has the potential to help to resolve the mechanism of transcriptional activation of Wnt target genes. The manuscript addresses an important mechanistic detail of the TCF/b-catenin complex, concerning the addition of the ChiLS complex and how the Wnt "enhanceosome" may be assembled. The dominant model on TCF/b-catenin mediated transcription is the "chain of co-activator" model, which posits that TCF recruits β-catenin that in turn recruits the BCL9 and Pygo complex and other co-activators. The authors have challenged this model through a series studies, including the eLife 2015 paper, and they have argued that the BCL9, Pygo and the new ChiLS complex is preassembled on Wnt enhanceosome to recruit β-catenin. Understanding the mechanism concerning TCF/β-catenin is of general interest and important so the paper in principle is worthy of publication in eLife. However, there are a number of aspects of the work which need to be strengthened to make a compelling case. A few additional experiments should allow the authors to provide the supporting evidence to solidify this model and the authors need to modify the text and discussion around their model where the data are inconsistent. We realize it may take time to conduct the two main experiments suggested but we are willing to allow that time to strengthen this story.

We were pleased that the referees found our study of interest, and stimulated by their challenging of our Wnt enhanceosome model and their generally constructive critique. We thus decided to design and conduct additional experiments to further test our conclusions and proposals, taking up the editors’ offer of a substantial extension of our revision deadline. In a nutshell, our additional work includes (1) in-depth analysis of transcriptional Wg responses in imaginal discs from larvae bearing our *lgs* truncation allele, (2) comparative BioID for PYGO2 as well as PYGO2 knock-out by CRISPR/Cas9, and (3) coIP assays to complement our evidence from BioID. We believe that our new results provide compelling support for the enhanceosome model. They also strengthen our insights into the general components of the Wnt enhanceosome that are not dedicated to Wnt signaling, such as Groucho/TLE, CBP/p300 and the subunits of the BAF complex (in particular Osa/ARID1), which broadens the interest of our study.

Inevitably though, our additional work (3 new main figures, with 11 panels total; plus 6 new supplemental figures, with 12 panels total) nearly doubled the data described in our manuscript, and we were thus compelled to recast it as a Research Article. This also allowed us more space to explain more clearly the rationales underlying our approaches, and to outline how our results advance and revise current models of Wnt-dependent transcription switches (including our own; see below). Neither of this was possible in our original submission as a Research Update, because of length restrictions. We hope the referees will agree with us that the new format is appropriate for our revised manuscript.

*The specific points to be addressed are:*

*1) The authors do not discuss some inconsistencies of their data with current models of Wnt signaling. They need to provide a more comprehensive interpretation of their data and how it might be reconciled with their previous studies. If the C-terminal region of Bcl9 proteins functions by binding the repressor TLE, one would expect deleting this region to enhance basal expression of Wnt target genes, rather than reducing their activation by Wnt. Analysis of Wg target gene expression in the lgs^2-8^ mutant might help to clarify this issue.*

In our revised manuscript, we discuss our data in the context of current views of Wnt-dependent transcription switches, adding several references to refer to results published by others on this topic, much of which is consistent with our new evidence (or might be re-interpreted in the light of it). For example, we explain that Groucho/TLE is a bimodal regulator, not only conferring silence on the enhanceosome in the absence of signaling, but also earmarking it (together with other core components) for subsequent Wnt induction (Discussion, fifth paragraph): owing to its multivalent interactions with TCF, ChiLS and the Legless/BCL9 C- terminus, Groucho/TLE assumes the role of a cornerstone within the enhanceosome (see our refined model in new Figure 7). As such, it may well be crucial for enhanceosome function (presumably by contributing to assembly and/or cohesion of the complex). It seems plausible that removing one of these interactions (e.g. by deleting its binding site in the Legless/BCL9 C-terminus) would compromise enhanceosome function and attenuate Wnt responses. We also point out in our revised manuscript (Discussion, last paragraph) that this positive role of Groucho/TLE within the Wnt enhanceosome is similar to its positive role in estrogen receptor signaling, as recently discovered by the Carroll group (Holmes et al., 2012): these authors showed that TLE1 promotes the interaction of estrogen receptor with chromatin and facilitates its function in stimulating proliferation of breast cancer cells.

Furthermore, we describe our BioID evidence (subsection “β-catenin triggers a Wnt-dependent apposition of the BCL9/B9L C-terminus to TCF”) that the CBP/p300 co-activator associates with the Wnt enhanceosome prior to Wnt signaling (revised Figure 4AB, D), possibly via direct interaction with Legless/BCL9, as indicated by RIME (new Figure 4).

Although we have not mapped this interaction, the best candidate for the CBP/p300 binding site in Legless/BCL9 is its C-terminus (given that this C-terminus contains three additional conserved sequence blocks, HD4-6, whose putative ligands have yet to be discovered). If so, this would further explain the positive role of this C-terminus, given that CBP/p300 is a well-established co-activator of transcription.

Interestingly, like Groucho/TLE, CBP/p300 is a bimodal regulator, acting activating or repressive depending on context, as shown by the Cadigan group and ourselves (Waltzer et al., 1998; Li et al., 2007). Furthermore, our BioID data indicated that both bimodal regulators are constitutively associated with the enhanceosome (Figure 4). Indeed, CBP has been found to be active in the absence of Wnt signaling, e.g. towards dTCF (Waltzer et al., 1998), or towards histones, as shown by the Cadigan group (Parker et al., 2008) who detected dCBP-dependent histone acetylation associated with Wg-responsive enhancers prior to Wg-dependent transcription (cited in the fourth paragraph of the Discussion).

Importantly, we have extended our phenotypic analysis of *lgs2-8* mutant larvae and flies (new Figure 2; new Figure 2—figure supplement 2C), and we have analyzed multiple Wg target genes in different imaginal discs of these mutants, including a Wg-responsive leg enhancer from *dpp (dpp.lacZ*) (new Figure 2; new Figure 2—figure supplement 2A, B). This additional work provides compelling evidence that the deletion of the C-terminus of Legless invariably produces *wg*-like effects (also mimicking attenuated *pygo* function). This supports our conclusion that the C-terminal portion of the protein, downstream of its Pygo-Armadillo adaptor sequences, is essential for promoting transcriptional Wg responses during fly development.

*It is also not clear why the Chip heterozygous phenotype, which looks like a loss of wg function, should be suppressed by removing positively acting components of the complex such as arm and lgs, as well as the negatively acting component gro.*

Genetic interactions caused by heterozygosity of genes (which reduces their expression levels by half, at least in *Drosophila*) are notoriously difficult to interpret. This gets worse if the proteins in question are components of a multiprotein complex, because it is hard to predict how reducing the expression levels of individual components will affect complex assembly, especially if the structure of this complex and its stoichiometry are poorly defined.

Nevertheless, in the revised figure legendof Figure 6—figure supplement 2, we provide what we consider a plausible explanation for the observed genetic interactions. Namely, it is well-established that the wing margin defects caused by halving the dose of *chip* means that this protein is severely limiting in wing discs (as discussed, and also exploited, extensively by the Dale, Cohen and Segal groups and others; see Fiedler et al., 2015).

This haploinsufficiency phenotype caused by *chip*/+ should therefore be exacerbated by lowering expression of its obligatory partner SSDP. It might be ameliorated by lowering the dose of enhanceosome components that, at their normal expression level, potentially sequester the limiting amounts of Chip by direct binding (e.g. Groucho, Pygo and Legless).

*The absence of any Runx protein in the BCL9 and B9L biotinylation assays argues against such a protein being a central part of the Wnt enhanceosome as proposed by Fiedler et al.*

Absolutely, and we now state this explicitly (Discussion, seventh paragraph). We offer the following (in our view plausible) explanation for this finding: namely, the role of RUNX in promoting Wnt enhanceosome function in the *Drosophila* embryo (as uncovered by our study of Wg-responsive embryonic midgut enhancers; Fiedler et al., 2015) is reminiscent of its pioneering functions in other uncommitted cell lineages, e.g. in the hematopoietic system. However, HEK293 cells are epithelial cells, and our data from BioID and Wnt reporter assays clearly show that the Wnt enhanceosome is fully functional in these cells without RUNX (indeed, we could not find any evidence for RUNX protein expression in these cells). A corollary is that RUNX functions in a hit-and-run fashion, to help set up the Wnt enhanceosome on Wnt-responsive enhancers in uncommitted cell lineages; however, once assembled (e.g. in epithelial cells), RUNX is no longer required for enhanceosome- mediated ON/OFF switching. Hit-and-run is a well-established concept in other contexts of transcriptional control during development, e.g. in setting up Polycomb-repressive complexes in the early *Drosophila* embryo.

*The observation that Wnt signaling is necessary for the interaction of Lgs with β-catenin and TCF proteins and does not reduce its association with TLE proteins does not support a model in which the enhanceosome is present at TCF-bound enhancers in the absence of Wnt signaling.*

This was our most intriguing BioID result – namely the Wnt-inducible labeling of TCF factors by both BCL9-BirA* and B9L-BirA*. We proposed in our initial submission that this reflects a β-catenin-dependent increase in proximity between TCF and the BCL9/B9L C- terminus (which bears the BirA* tag; see also below, point 3). We now provide two new strands of evidence to support this notion: firstly, we use coIP assays to show that the association of BCL9/B9L with TCF is similarly stable before and after Wnt stimulation, like its association with Pygo, while its association with β-catenin depends on Wnt stimulation (new Figure 4—figure supplement 3). Secondly, we have conducted BioID with PYGO2-BirA* under identical conditions (new Figure 4—figure supplement 1), which revealed no new Figure 4 significant change in the labeling of TCF factors by this bait (), in contrast to the labeling of TCF factors by the BCL9/B9L baits which is strongly Wnt- inducible, especially in the case of B9L-BirA* (13-30x; Figure 4). Apart from these factors, the hits identified by PYGO2-BirA* are remarkably similar to those identified by the BCL9/B9L baits (summarized in new Figure 4—figure supplement 2), showing constitutive labeling of virtually all enhanceosome components. Notable exceptions are β-catenin and CBP/p300 whose labeling is Wnt-dependent or Wnt-stimulated, respectively, as in the case of the BCL9/B9L baits (see above).

As in the case of the BCL9/B9L baits, the list of constitutively labeled components by PYGO2-BirA* includes TLE3 and TLE1, underscoring our conclusion that these co- repressors remain associated with the Wnt enhanceosome even during Wnt signaling. Although this is against the prevailing view of Groucho/TLE detaching from TCF factors as a result of their binding to β-catenin, but to our knowledge, this has never been explicitly tested (e.g. by ChIP). Unfortunately, the Groucho antiserum previously used for ChIP (e.g. by the Jennings group) and initially produced by Kharchenko et al. (2011) is no longer available. However, this notion of competition between Groucho/TLE and Armadillo/β-catenin was seriously challenged recently by the discovery of Weis and Waterman and co-workers that TLE and β-catenin can bind to TCF simultaneously (Chodaparambil et al., 2014), as we point out in our Introduction.

PYGO2-BirA* also labels the nuclear co-receptor components, like BCL9-BirA* and B9L- BirA*. This strengthens the link between the Wnt enhanceosome and nuclear hormone receptor signaling, as discussed in the last paragraph of the Discussion. In this paragraph, we also cite work by the Carroll group who found a strong association of TLE1 with estrogen receptor-occupied loci in breast cancer cells, and a positive role of TLE1 in promoting the function of this nuclear hormone receptor (Holmes et al., 2012). As mentioned above, this is reminiscent of its role in promoting Wnt enhanceosome function and Wnt responses.

Finally, in the second paragraph of the subsection “β-catenin triggers a Wnt-dependent apposition of the BCL9/B9L C-terminus to TCF”, we refer to work by Parker et al. (2002) who provided ChIP-based evidence that, in *Drosophila*, the association of dTCF with target enhancers is increased by Wg signaling. This is not inconsistent with out BioID data from the BCL9/B9L baits which show slightly increased labeling of most Wnt enhanceosome components, although this increase is not seen with the PYGO2 bait. In any case, TCF factors clearly associate with target enhancers in the absence of Wnt signaling in HEK293 cells, as shown by Frietze et al. (2012) (cited in the aforementioned paragraph). Whether or not the binding of β-catenin to Legless/BCL9 and/or TCF results in a stabilization of these interactions will require future testing.

*2) The Authors employed BioID (using BirA*) followed by mass spectrometry to identify BCL9 (and the related B9L) binding partners (TLE, ChiLS, and nuclear receptor co-activators). They went one step further and combined BioID and spectral counting as a (semi) quantitative measurement for determination of protein interactions with and without Wnt. It is not clear to me whether these methods offer reliable quantification. For example, spectral counts for LDB1 and SSDP in Figure 3 appear to contradict the IP results in Figure 3, in particular with regard to the HD1 deletion mutant. It would be highly desirable that additional approaches can back up these arguments (derived from BioID).*

In the revised manuscript, we state more clearly that coIP assays assess (relatively stable) associations between proteins, while BioID assesses primarily their proximity (even if this reflects weak binding affinities). We have added an introductory paragraph on BioID (subsection “B9L and BCL9 are constitutively associated with the Wnt enhanceosome”, first paragraph), explaining how the spectral counts obtained by this approach can be taken as (semi)quantitative measurements of proximity, which can provide insight into the spatial organization of large protein complexes. This was demonstrated in the original Roux et al. (2012) paper by comparing biotin labeling efficiencies of interactors of Laminin A tagged with N-terminal or C-terminal BirA*. Proximity-dependent labeling efficiencies have since been documented by others, and we cite a compelling example of BioID data obtained by the Stearns group on components of the human centrosomal complex (Firat-Karalar et al., 2014). In the same paragraph, we also mention other factors that can affect the efficiency of biotin labeling by the bait, but these are less relevant for the comparison of hits obtained under different conditions (with or without Wnt), or of hits labeled by different baits under the same conditions (BCL9/B9L versus PYGO2).

Finally, as mentioned above, we have complemented our evidence from BioID with coIP assays, which revealed a relatively stable association between TCF4 and B9L in unstimulated cells, which is only slightly increased upon stimulation (see above; new Figure 4—figure supplement 3).

*One of the key arguments of the authors' model is that Pygo/BCL9 and associated ChiLS are preassembled on chromatin prior to Wnt stimulation and recruits b-catenin to TCF upon Wnt signaling. The authors should demonstrate this important issue using other techniques, such as CHIP.*

We are unable to conduct ChIP experiments since ChIP-grade antisera for these proteins are not (or no longer) available. Instead, we have conducted coIP experiments, and also comparative BioID labeling by PYGO2-BirA*, to complement and consolidate our conclusions based on the BCL9/B9L BioID baits (as detailed above).

*3) The NMR results showed that HD3 directly interacts with LDB1/SSDP, and HD3 deletion and W472A mutants reduced BCL9-LDB1 coIP. But HD1 appeared to have a more critical role as its deletion completely abolished BCL9-LDB1 coIP. The authors suggested, reasonably, that HD1 binding to Pygo, which interacts with LDB1, has a critical role in this multivalent interaction. The authors should be able to demonstrate this point directly using Pygo knockdown or knockout (via CRISPR).*

We have done this, by knocking out PYGO2 by CRISPR/Cas9 (new Figure 5—figure supplement 2). The results (shown in new =Figure 5—figure supplement 3) support our notion of Pygo’s critical role in promoting the interaction between LDB1 and BCL9/B9L.